# Myristic acid as a checkpoint to regulate STING-dependent autophagy and interferon responses by promoting N-myristoylation

Mutian Jia [1,4], Yuanyuan Wang[1,4], Jie Wang[1], Danhui Qin[1], Mengge Wang[1], Li Chai [1], Yue Fu[1,2], Chunyuan Zhao[1,3], Chengjiang Gao [1], Jihui Jia[1] & Wei Zhao [1] ✉

Stimulator of interferon gene (STING)-triggered autophagy is crucial for the host to eliminate invading pathogens and serves as a self-limiting mechanism of STING-induced interferon (IFN) responses. Thus, the mechanisms that ensure the beneficial effects of STING activation are of particular importance. Herein, we show that myristic acid, a type of long-chain saturated fatty acid (SFA), specifically attenuates cGAS-STING-induced IFN responses in macrophages, while enhancing STING-dependent autophagy. Myristic acid inhibits HSV-1 infection-induced innate antiviral immune responses and promotes HSV-1 replication in mice in vivo. Mechanistically, myristic acid enhances N-myristoylation of ARF1, a master regulator that controls STING membrane trafficking. Consequently, myristic acid facilitates STING activation-triggered autophagy degradation of the STING complex. Thus, our work identifies myristic acid as a metabolic checkpoint that contributes to immune homeostasis by balancing STING-dependent autophagy and IFN responses. This suggests that myristic acid and N-myristoylation are promising targets for the treatment of diseases caused by aberrant STING activation.

The DNA-sensing receptor cyclic guanosine monophosphate (GMP)-adenosine monophosphate (AMP) (cGAMP) synthase (cGAS), and its downstream signaling effector stimulator of interferon gene (STING) serve as fundamental elements of the immune system to detect microbial DNA or self-DNA in the cytoplasm, and then initiate immune responses for host defense[1–4]. Upon DNA binding, cGAS produces the second messenger cGAMP, which binds to and activates STING[5–7]. Activated STING translocates from the endoplasmic reticulum (ER) to the Golgi apparatus via the ER-Golgi intermediate compartment (ERGIC), where STING forms a complex with TANK-binding kinase 1 (TBK1) and interferon regulatory factor 3 (IRF3) to initiate the expression of type I interferons (IFNs)[2]. STING trafficking also triggers

the formation of autophagosomes via a non-canonical mechanism of autophagy. STING-dependent autophagy is a primordial function of the cGAS pathway and plays important roles in the clearance of cytosolic DNA and invading viruses[8,9]. Meanwhile, both STING within autophagosomes and STING from the Golgi traffic to the lysosome[3,4,8,10], which triggers STING protein degradation and serves as a self-limiting mechanism of the cGAS-STING pathway. Therefore, the balance between STING-dependent autophagy and IFN responses is vital for eliminating invading pathogens and maintaining immune homeostasis.

Lipid metabolism such as synthesis and oxidation of fatty acids undergoes profound alterations during infection and controls the

[1]Key Laboratory for Experimental Teratology of the Chinese Ministry of Education, and Key Laboratory of Infection and Immunity of Shandong Province, School of Basic Medical Science, Qilu Hospital, Cheeloo College of Medicine, Shandong University, Jinan, Shandong, China. [2]Department of Physiology and Pathophysiology, School of Basic Medical Science, Cheeloo College of Medicine, Shandong University, Jinan, Shandong, China. [3]Department of Cell Biology, School of Basic Medical Science, Cheeloo College of Medicine, Shandong University, Jinan, Shandong, China. [4]These authors contributed equally: Mutian Jia, Yuanyuan Wang. ✉e-mail: wzhao@sdu.edu.cn

status of immune cell activation[11]. Fatty acid synthesis (FAS) is an integral part of macrophage activation and is considered essential for membrane biogenesis[11,12]. Fatty acid oxidation not only provides energy for tolerogenic immune cells (such as memory T cells and regulatory T cells) but also regulates the phenotype and function of macrophages[13–16]. Our previous study showed that viral infection induced lipid peroxidation, which attenuated cGAS-triggered innate immunity by promoting STING carbonylation[17]. Saturated fatty acids (SFAs) are used as energy sources and structural elements for protein modification[18]. Palmitic acid and myristic acid are two common long-chain SFAs that can modify proteins (termed palmitoylation and myristoylation, respectively), thereby altering their functions. Intracellular SFA-dependent palmitoylation of myeloid differentiation factor 88 (MyD88) is required for Toll-like receptor (TLR) mediated inflammation[19]. The palmitoylation of STING is vital for its anchoring to the Golgi and facilitates type I IFN production[20]. Myristic acid, a 14-carbon straight-chain SFA, can be attached to N-terminal glycine residues by N-myristoyltransferase (NMT) to give rise to another modification, termed N-myristoylation. N-myristoylation plays an important role in cell signaling and allows for the dynamic interactions of proteins with cell membranes or lipid rafts[21,22]. For example, the lysosomal translocation and activation of AMP-activated protein kinase (AMPK) require its N-myristoylation[23]. The concentrations of intracellular free fatty acids, including myristic acid, are higher in rheumatoid arthritis (RA) T cells and increase AMPK activation and

proinflammatory T cell phenotypes[24]. N-myristoylation of TRIF-related adapter molecule (TRAM) targets it to the plasma membrane and facilitates TLR4 signal transduction[25]. However, the potential roles of myristic acid and N-myristoylation in cGAS-STING-dependent innate immunity remain unknown.

In this study, we showed that myristic acid attenuates cGAS- and STING-dependent expression of type I IFNs by enhancing N-myristoylation. Myristic acid enhances N-myristoylation of the GTPase ADP-ribosylation factor 1 (ARF1), a master regulator that controls STING membrane trafficking[8,26], and facilitates STING-dependent autophagy degradation of the STING/TBK1 complex. Our work clarifies how myristic acid, as a metabolic checkpoint, balances STING-dependent IFN responses and autophagy. In addition, our study uncovered a mechanism by which N-myristoylation controls STING activation and suggests a promising approach for modulating STING-dependent immunopathologies.

## Results

### Myristic acid inhibits cGAS-dependent antiviral innate immunity

To investigate the potential regulatory roles of SFAs in the cGAS pathway in macrophages, we first analyzed the expression pattern of SFAs during the activation of cGAS signaling in mouse primary peritoneal macrophages (PMs). Herpes simplex virus-1 (HSV-1, a DNA virus recognized by cGAS) infection decreased the cellular level of myristic acid (C14:0), with no effects on other SFAs (Fig. 1a). Next, the effects of

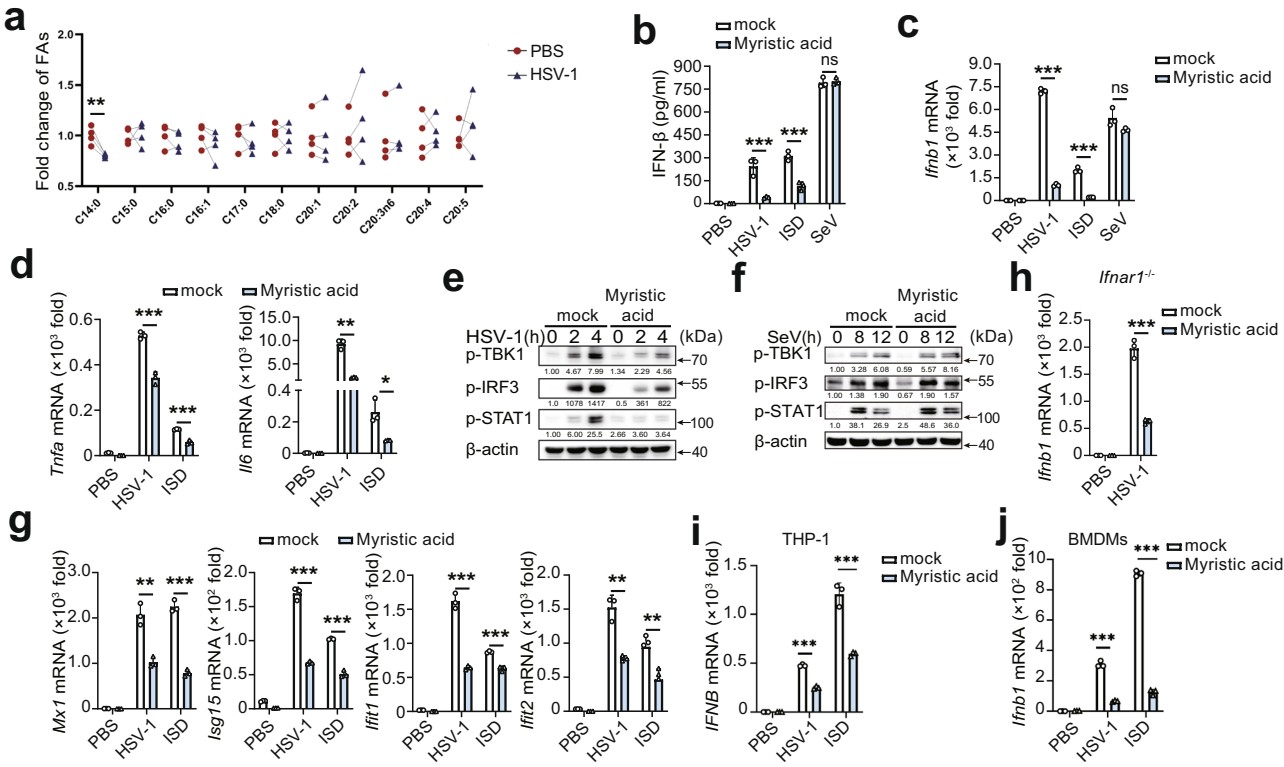

**Fig. 1 | Myristic acid inhibits cGAS-dependent antiviral immune responses. a** Fold change of saturated fatty acids (SFAs), determined by gas chromatography in primary mouse peritoneal macrophages (PMs). Statistical significance was determined by paired two-sided multiple Student's t-tests, n = 4 biologically independent samples. **b−d** Enzyme-linked immunosorbent assay (ELISA) analysis of interferon (IFN)-β secretion (**b**) and quantitative polymerase chain reaction (qPCR) analysis of *Ifnb1, Tnfa,* or *Il6* mRNA level (**c** and **d**) in mouse PMs pretreated with solvent (mock) or myristic acid, plus stimulation as indicated, n = 3 samples examined over 3 independent experiments. **e, f** Immunoblot assays of p-TBK1, p-IRF3, and p-STAT1 in PMs pretreated with solvent (mock) or myristic acid, and then infected with herpes simplex virus 1 (HSV-1) or Sendai virus (SeV). **g** qPCR analysis

of *Mx1, Isg15, Ifit1,* and *Ifit2* mRNA level in PMs pretreated with solvent (mock) or myristic acid, and then infected with HSV-1 or transfected with IFN-stimulating DNA (ISD). **h** qPCR analysis of *Ifnb1* in *Ifnar1*-deficient PMs pretreated with solvent (mock) or myristic acid, and then infected with HSV-1, n = 3 samples examined over 3 independent experiments. **i, j** qPCR analysis of *IFNB* mRNA level in THP-1 (**i**) and bone marrow-derived macrophages (BMDMs) (**j**), n = 3 samples examined over 3 independent experiments. Statistical significance was determined by unpaired two-sided multiple Student's t-tests in (**b−d**) and (**h−j**). Data are shown as mean ± standard deviation (SD) or typical photographs and are representative of three biological independent experiments with similar results. **P < 0.01, ***P < 0.001. Source data is provided in the Source data file.

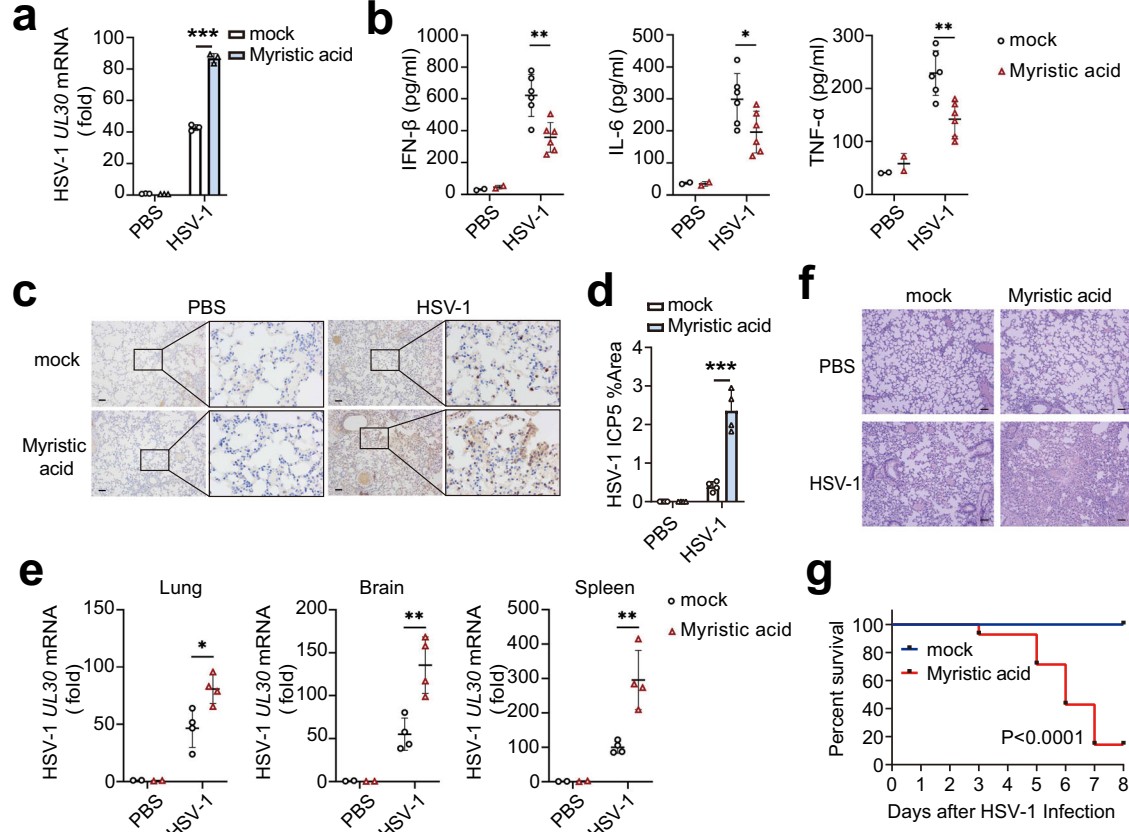

**Fig. 2 | Myristic acid inhibits immune responses against HSV-1 in vivo. a** Quantitative polymerase chain reaction (qPCR) analysis of HSV-1 UL30 mRNA level in peritoneal macrophages (PMs) pretreated with solvent (mock) or myristic acid, and then infected with HSV-1, $n = 3$ samples examined over 3 independent experiments. **b–g** C57BL/6J mice were pretreated with myristic acid, and then infected with HSV-1 ($2 \times 10^7$ p.f.u. per mouse) by intraperitoneal injection. Serum levels of IFN-β, IL-6, and TFN-α were analyzed by enzyme-linked immunosorbent assay (ELISA) (PBS group, $n = 2$; mock HSV-1 group, $n = 6$) (**b**). Immunohistochemistry (IHC) analysis of HSV-1 protein ICP5. Scale bar, 100 µm (**c** and **d**). qPCR analysis of HSV-1 UL30 mRNA level in lung, brain, and spleen (PBS group, $n = 2$; mock HSV-1 group, $n = 4$) (**e**). Haematoxylin and eosin staining of lung tissue sections. Scale bar, 100 µm (**f**), data are shown as typical photographs and are representative of biological samples (PBS group, $n = 2$; mock HSV-1 group, $n = 4$). Kaplan–Meier method was used to evaluate survival curves ($n = 14$ per group) (**g**). Statistical significance was determined by unpaired two-sided multiple Student's t-tests in (**a**), (**b**), (**d**), and (**e**) or the log-rank Mantel–Cox test in (**g**). Data are shown as mean ± standard deviation (SD). *$P < 0.05$, ***$P < 0.001$. Source data is provided in the Source data file.

several major SFAs on cGAS activation were examined in HSV-1-infected PMs. Myristic acid most significantly attenuated HSV-1 infection-induced *Ifnb1* mRNA expression in these major SFAs, including palmitic acid, linoleic acid, and stearic acid (Fig. S1a). Notably, myristic acid did not show toxic effects in PMs, even at doses twice as high as those used (Fig. S1b). Similarly, myristic acid treatment also inhibited HSV-1 infection- and interferon-stimulating DNA- (ISD, which can be recognized by cGAS) induced IFN-β secretion and mRNA expression (Fig.1b, c). However, Sendai virus- (SeV, an RNA virus recognized by RIG-I) induced IFN-β secretion and mRNA expression was not affected by myristic acid treatment (Fig. 1b, c). In addition, myristic acid treatment decreased HSV-1-infection- and ISD-induced the expression of tumor necrosis factor-α (*Tnfa*), and interleukin-6 (*Il6*) (Fig. 1d). Overall, these data indicate that myristic acid selectively attenuates cGAS signaling.

cGAS activation triggers the downstream activation of TBK1 and IRF3, initiating the expression of type I IFNs, which further induces the production of IFN-stimulated genes (ISGs such as Mx1, ISG15, IFIT1, and IFIT2) through the Janus kinase (JAK)–signal transducer and activator of transcription (STAT) pathway[1,27]. Myristic acid treatment suppressed both HSV-1 and ISD, but not SeV, induced phosphorylation of TBK1, IRF3, and STAT1 (Figs. 1e, f and S1c). Consequently, the expression of ISGs and C-X-C motif chemokine 10 (*Cxcl10*) was also suppressed by myristic acid (Figs. 1g and S1d).

IFN-β binds to type I IFN receptor (IFNAR), and then activates the JAK–STAT1 pathway, which is vital for the feedback enhancement of type I IFN expression. Myristic acid treatment led to the reduction of *Ifnb* mRNA expression in *Ifnar1*-deficient macrophages following HSV-1 infection (Fig. 1h), which suggests that myristic acid functions upstream of IFNAR activation. Furthermore, myristic acid treatment inhibited HSV-1 infection- and ISD-induced IFN-β mRNA expression in THP-1 cells and bone marrow-derived macrophages (BMDMs) (Fig. 1I, j). These data further confirmed that myristic acid specifically attenuates cGAS-dependent innate immune responses.

Myristic acid treatment enhanced HSV-1 replication in PMs (Fig. 2a). We next investigated the physiological and pathological relevance of the regulatory effects of myristic acid on viral infection in vivo. Myristic acid-treated mice produced less IFN-β, IL-6, and TNF-α than that of control mice during HSV-1-infection (Fig. 2b). The amount of HSV-1 in the lung, brain, and spleen were much higher in myristic acid-treated mice than in control mice at 36 h after viral infection (Fig. 2c–e). More infiltration of inflammatory cells was observed in the lungs of myristic acid-treated mice after HSV-1 infection (Fig. 2f). Furthermore, myristic acid-treated mice were more susceptible in the survival assays upon HSV-1 infection than the control mice (Fig. 2g). Overall, these data suggest that myristic acid is an endogenous suppressor of cGAS-dependent antiviral innate immunity.

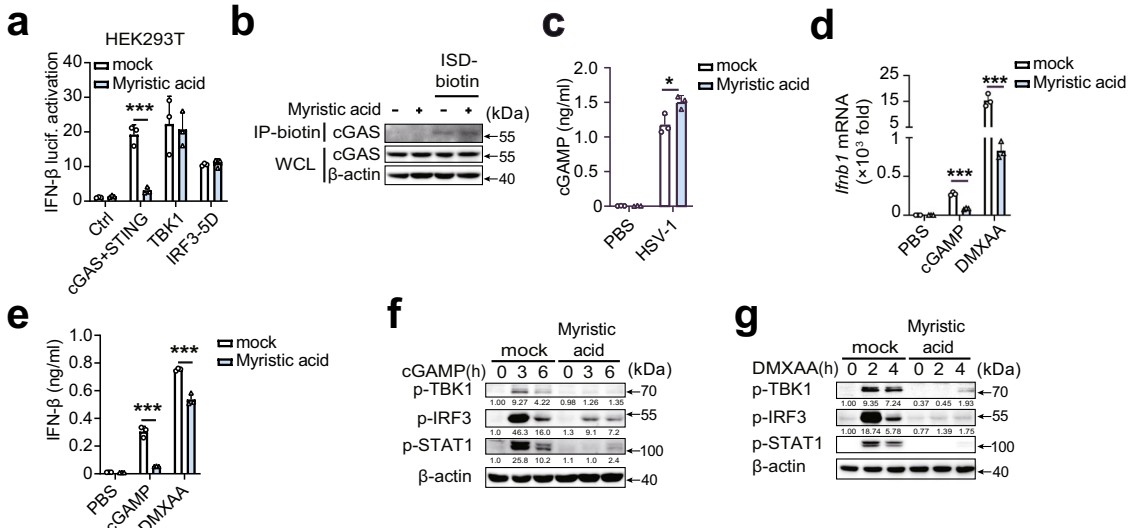

**Fig. 3 | Myristic acid inhibits STING activation. a** Luciferase activity analysis of HEK293T cells transiently transfected with interferon (IFN)-β reporter plasmid and adapter plasmids as indicated, *n* = 3 samples examined over 3 independent experiments. **b** In vitro pull-down assay analysis of IFN-stimulating DNA (ISD)-biotin binding to cGAS. **c** Enzyme-linked immunosorbent assay (ELISA) analysis of cyclic adenosine monophosphate (AMP)–guanine monophosphate (GMP) (cGAMP) production in peritoneal macrophages (PMs) treated with solvent (mock) or myristic acid, followed by herpes simplex virus 1 (HSV-1) infection, *n* = 3 samples examined over 3 independent experiments. **d**–**g** Quantitative polymerase chain reaction (qPCR) analysis of *Ifnb1* mRNA level (**d**), ELISA analysis of IFN-β secretion (**e**), *n* = 3 samples examined over 3 independent experiments, and immunoblot assays of p-TBK1, p-IRF3, and p-STAT1 (**f**, **g**) in PMs pretreated with solvent (mock) or myristic acid, following stimulation with cGAMP or 5,6-dimethylxanthenone-4-acetic acid (DMXAA). Statistical significance was determined by unpaired two-sided multiple Student's t-tests in (**a**) and (**c**–**e**). Data are shown as mean ± standard deviation (SD) or typical photographs and are representative of three biological independent experiments with similar results. *P < 0.05, ***P < 0.001. Source data is provided in the Source data file.

## Myristic acid inhibits STING activation

To clarify the mechanism by which myristic acid attenuates cGAS-dependent signaling, we first performed luciferase reporter assays. cGAS-STING-induced IFN-β luciferase activation was considerably inhibited in myristic acid-treated HEK293T cells, whereas no differences in TBK1- and IRF3-induced IFN-β luciferase activation were observed (Fig. 3a), suggesting that myristic acid may target cGAS or STING. However, myristic acid had no effect on the binding activity of cGAS with ISD (Fig. 3b). Myristic acid also did not decrease cGAMP production during HSV-1 infection (Fig. 3c). These results suggested that myristic acid may target the downstream of STING activation. Next, we further detected the function of myristic acid in cGAMP and STING agonist 5,6-dimethylxanthenone-4-acetic acid (DMXAA) induced STING activation. Myristic acid treatment markedly inhibited cGAMP- and DMXAA-induced IFN-β mRNA expression and secretion (Fig. 3d, e), and phosphorylation of TBK1, IRF3, and STAT1 (Fig. 3f, g). Collectively, these data indicate that myristic acid targets STING, but not cGAS.

## Myristic acid facilitates STING-dependent autophagy

Following HSV-1 infection, myristic acid treatment inhibited STING protein expression in a dose-dependent manner (Fig. 4a). Interestingly, myristic acid treatment also inhibited TBK1 and IRF3 protein expression in HSV-1-infected PMs (Fig. 4a), with no effects on the mRNA expression of *sting*, *tbk1*, and *irf3* (Fig. S2). In cGAMP-activated HEK293T cells, myristic acid treatment also attenuated STING-Myc protein expression (Fig. 4b). Treatment with 3-methyladenine (3-MA) or chloroquine (inhibitors of the autophagy-lysosome degradation pathway), but not the proteasome inhibitor MG132, reversed the myristic acid-induced STING-Myc protein decrease (Fig. 4b). Furthermore, chloroquine or 3-MA treatment also reversed the inhibitory effect of myristic acid on STING, TBK1, and IRF3 protein expression in HSV-1-infected PMs (Fig. 4c, d). STING activation induces the formation of the STING-TBK1 complex in the Golgi apparatus. STING V147L and

N154S mutants were constitutively localized to the Golgi and activated downstream signaling. Myristic acid attenuated the protein expression of STING V147L and N154S mutants (Fig. 4e). Myristic acid treatment inhibited cGAMP-induced localization of STING at the Golgi (Fig. 4f). However, in STING-deficient PMs, myristic acid did not decrease TBK1 protein expression following HSV-1 infection or cGAMP stimulation (Fig. 4g, h). Furthermore, myristic acid treatment enhanced HSV-1 replication in both wild-type PMs and mouse embryonic fibroblasts (MEFs), but not in *Sting*-deficient PMs and MEFs (Fig. 4I, j). Overall, these data indicate that myristic acid facilitates the autophagy degradation of the STING-TBK1 complex in a STING-dependent manner.

The formation of STING-containing ERGIC is crucial for LC3 lipidation, autophagosome biogenesis, and subsequent lysosomal degradation[8]. Myristic acid treatment promoted HSV-1 infection-induced STING-LC3 puncta formation and cGAMP-induced LC3-I to LC3-II conversion (Fig. 4k, l). Upon STING activation, a part of STING traffic from the ERGIC to the Golgi, where STING activates TBK1 and leads to induction of type I IFN, but other parts of STING at the ERGIC lead to the formation of autophagosomes[8]. Our results showed that myristic acid treatment inhibited cGAMP-induced localization of STING at the Golgi (Fig. 4f). These results suggest that myristic acid facilitates STING-triggered autophagy and promotes autophagy degradation of the STING complex.

## Myristic acid inhibits STING activation by enhancing N-myristoylation

Myristic acid can attach to the N-terminal glycine (Gly) residue of targets via a covalent amide bond, a process named protein N-myristoylation[21,22]. Free myristic acid must be esterified to coenzyme A-activated intermediate myristoyl-CoA catalyzed by NMTs onto the N-terminal glycine[28]. Myristic acid treatment induced high levels of protein N-myristoylation in PMs (Fig. S3a). Myristoyl-CoA treatment inhibited HSV-1-induced phosphorylation of TBK1 and IRF3, suggesting a regulatory role for N-myristoylation in the cGAS-STING pathway

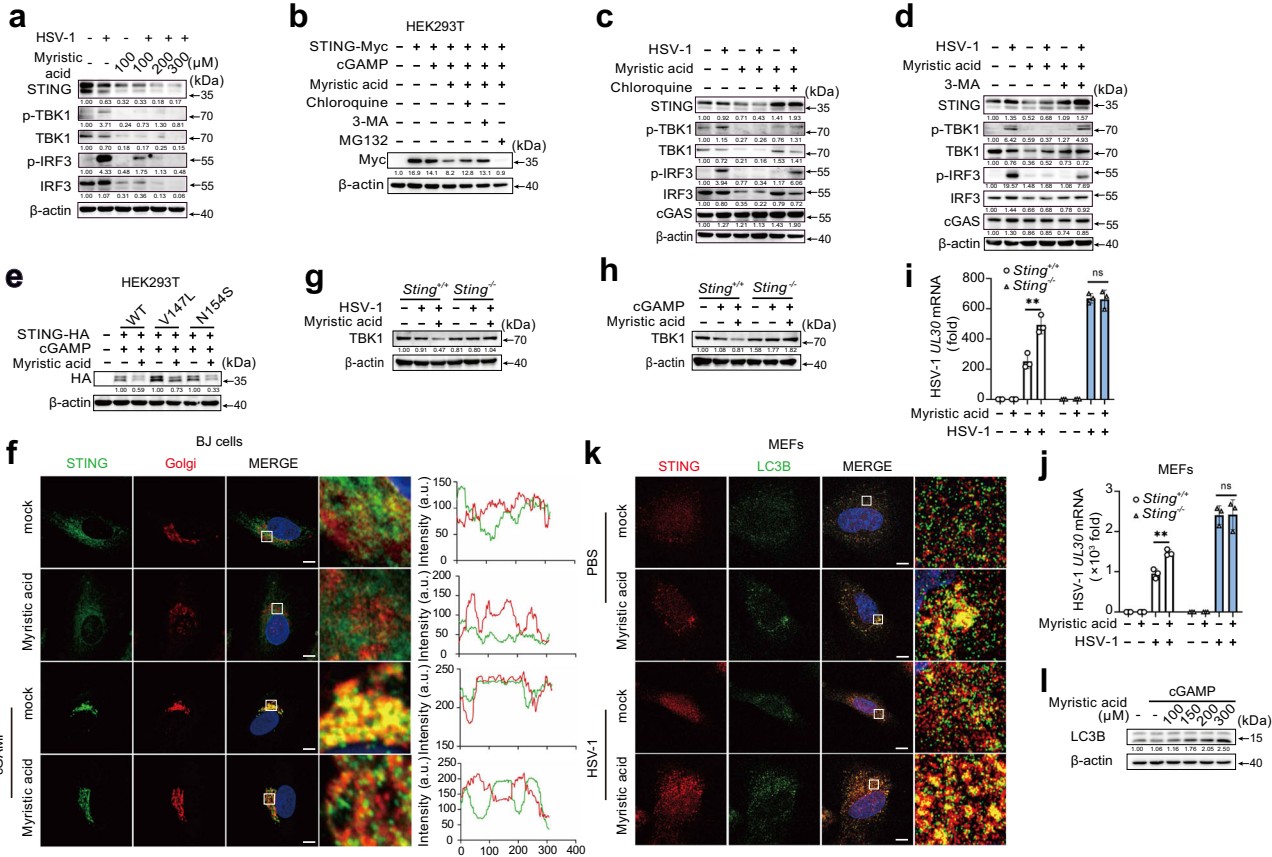

**Fig. 4 | Myristic acid promotes STING-dependent autophagy. a** Immunoblot analysis of indicated proteins in peritoneal macrophages (PMs) pretreated with increasing concentrations of myristic acid, following herpes simplex virus 1 (HSV-1) infection. **b** Immunoblot analysis of STING expression in STING-Myc transfected HEK293T cells pretreated with chloroquine, 3-methyladenine (3-MA), or MG132, and then treated with myristic acid, following cyclic adenosine monophosphate (AMP)–guanine monophosphate (GMP) (cGAMP) stimulation. **c**, **d** Immunoblot analysis of indicated proteins in PMs pretreated with chloroquine (**c**) or 3-MA (**d**), and then treated with myristic acid, followed by HSV-1 infection. **e** Immunoblot analysis of STING expression in HEK293T cells transfected with empty vector (ctrl), wild-type STING (WT), or STING mutants (V147L or N154S), and then treated with myristic acid, following cGAMP stimulation. **f** Confocal analysis of STING and Golgi in BJ cells pretreated with solvent (mock) or myristic acid and then stimulated with cGAMP. Scale bar, 10 μm. Intensity profiles of each line were quantified by ImageJ

software. **g**, **h** Immunoblot analysis of TBK1 expression in *Sting* or *Sting* PMs treated with myristic acid, following HSV-1 infection (**g**) or cGAMP stimulation (**h**). **I**, **j** Quantitative polymerase chain reaction (qPCR) analysis of HSV-1 UL30 mRNA level in peritoneal macrophages (PMs) (**i**) or mouse embryonic fibroblasts (MEFs) (**j**) from *Sting* or *Sting* mice, pretreated with solvent (mock) or myristic acid, and then infected with HSV-1, *n* = 3 samples examined over 3 independent experiments. **k** Confocal analysis of STING and LC3B-GFP in MEFs pretreated with solvent (mock) or myristic acid and then infected with HSV-1. Scale bar, 10 μm. **l** Immunoblot analysis of LC3 in PMs pretreated with chloroquine and increasing concentrations of myristic acid, followed by cGAMP stimulation. Statistical significance was determined by unpaired two-sided multiple Student's t-tests in (**i**) and (**j**). Data are shown as mean ± standard deviation (SD) or typical photographs and are representative of three biological independent experiments with similar results. ***P < 0.001. Source data is provided in the Source data file.

(Fig. S3b). We then investigated whether endogenous myristic acid could regulate STING activation by N-myristoylation. 2-Hydroxytetradecanoic acid (2-HOM), DDD85646, and IMP-1088 are different inhibitors of myristoylation[29–31] and show no toxic effects in PMs (Fig. S4a). Treatment with 2-HOM, DDD85646, or IMP-1088 treatment enhanced HSV-1 infection-, cGAMP-, and DMXAA-induced *Ifnb1* mRNA expression (Fig. 5a–c), and DDD85646 treatment enhanced ISD- and DMXAA-induced IFN-β secretion (Fig. S4b). Similarly, 2-HOM or DDD85646 treatment enhanced IFN-β luciferase activation in DMXAA-stimulated 293-Dual hSTING-A162 cells (Fig. 5d). DDD85646 treatment promoted DMXAA-induced phosphorylation of TBK1 and IRF3 in PMs in a dose-dependent manner (Fig. 5e). Furthermore, myristic acid-mediated decrease in IFN-β expression could be partially reversed by DDD85646 or IMP-1088 treatment (Fig. 5f–h). Consistently, inhibition of N-myristoylation by DDD85646 reversed myristic acid-enhanced STING-dependent degradation of STING, TBK1, and IRF3 in PMs (Fig. 5i). Similar results were obtained in STING-overexpressing HEK293T cells (Fig. S4c). NMT inhibitor DDD85646

treatment inhibited cGAMP-induced LC3-I to LC3-II conversion in wild-type PMs, but not in STING-deficient PMs (Fig. 5j), suggesting NMT-induced N-myristoylation enhanced STING-mediated autophagy. To confirm the endogenous role of myristic acid in the regulation of STING pathway in vivo, we constructed the animal model of DDD85646 oral administration. DDD85646-treated mice produced higher IFN-β, IL-6, and TNF-α than that of control mice during HSV-1-infection (Fig. 5k). The replication of HSV-1 in the lung, brain, and spleen was reduced in DDD85646-treated mice than in control mice (Fig. 5l). Collectively, these data indicate that myristic acid facilitates STING-triggered autophagy degradation of the STING complex by enhancing N-myristoylation, thereby attenuating the cGAS-STING pathway.

N-myristoylation is catalyzed by N-myristoyltransferases (NMTs), including two isozymes, N-myristoyltransferase 1 (NMT1) and 2 (NMT2)[22], which possess distinct substrate specificity[32,33]. NMT1, but not NMT2, is ubiquitously expressed in immune cells and upregulated following HSV-1 infection (Fig. 6a, b). Next, we examined the potential

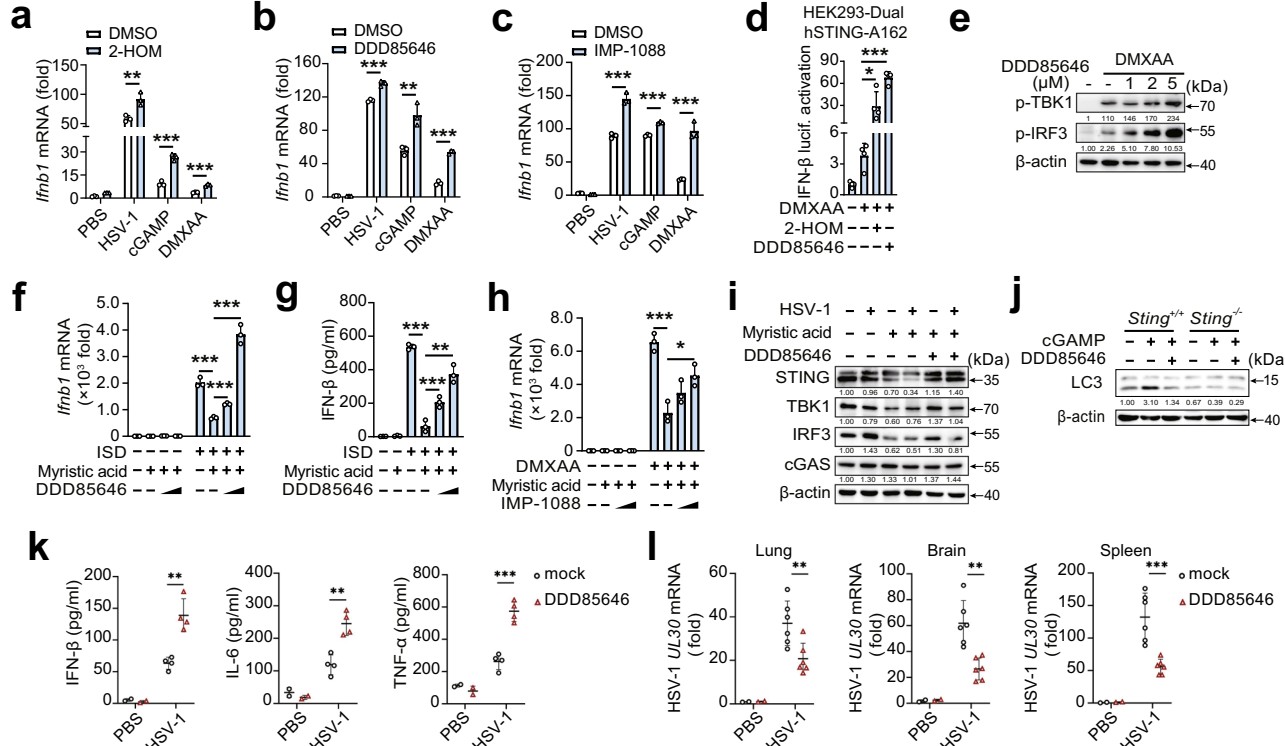

**Fig. 5 | Myristic acid inhibits cGAS-STING activation via facilitating N-myristoylation. a–c** Quantitative polymerase chain reaction (qPCR) analysis of *Ifnb1* mRNA in peritoneal macrophages (PMs) pretreated with DMSO, 2-HOM, DDD85646, or IMP-1088, following HSV-1 infection, cGAMP or 5,6-dimethyl-xanthenone-4-acetic acid (DMXAA) stimulation, $n = 3$ samples examined over 3 independent experiments. **d** Luciferase activity analysis of interferon (IFN)-β activation in HEK293-Dual hSTING-A162 cells pretreated with DMSO, 2-HOM, or DDD85646, followed by DMXAA stimulation, $n = 3$ samples examined over 3 independent experiments. **e** Immunoblot analysis of p-TBK1 and p-IRF3 in PMs pretreated with DMSO or DDD85646, and then stimulated with DMXAA. **f, g** qPCR analysis *Ifnb1* mRNA expression (**f**) or enzyme-linked immunosorbent assay (ELISA) (**h**) analysis of IFN-β secretion in PMs pretreated with DDD85646, and then treated with myristic acid, followed by ISD stimulation, $n = 3$ samples examined over 3 independent experiments. **h** qPCR analysis of *Ifnb1* mRNA in PMs pretreated with

IMP-1088, and then treated with myristic acid, followed by DMXAA stimulation, $n = 3$ samples examined over 3 independent experiments. **i** Immunoblot analysis of indicated proteins in PMs pretreated with DDD85646, and then treated with myristic acid, followed by HSV-1 infection. **j** Immunoblot analysis of LC3 in PMs from *Sting* or *Sting* mice pretreated with DDD85646, followed by cGAMP stimulation. **k, l** C57BL/6J mice were orally administered with DDD85646 and then infected with HSV-1 ($2 \times 10^{+/+-/7}$ p.f.u. per mouse) by intraperitoneal injection. Serum levels of IFN-β, IL-6, and TFN-α were analyzed by ELISA (PBS group, $n = 2$; mock HSV-1 group, $n = 4$) (**k**). qPCR analysis of HSV-1 UL30 mRNA level in lung, brain, and spleen (PBS group, $n = 2$; mock HSV-1 group, $n = 6$) (**l**). Statistical significance was determined by unpaired two-sided multiple Student's t-tests in (**a–d**), (**f–h**), and (**k, l**). Data are shown as mean ± standard deviation (SD) or typical photographs and are representative of three biological independent experiments with similar results. *$P < 0.05$, **$P < 0.01$, ***$P < 0.001$. Source data is provided in the Source data file.

role of NMT1 in cGAS-STING activation. NMT1 inhibited cGAS-STING-triggered IFN-β reporter activation (Fig. 6c). *Nmt1* knockdown substantially promoted HSV-1 infection-, cGAMP-, and DMXAA-induced *Ifnb1* expression in PMs (Fig. 6d–f). Moreover, *Nmt1* knockdown partially reversed the decrease in phosphorylation of TBK1 and IRF3 and *Ifnb1* mRNA expression in myristic acid-treated PMs (Fig. 6g, h). Overall, these data further confirm that myristic acid inhibits STING activation by regulating N-myristoylation.

### Myristic acid enhances ARF1 N-myristoylation and facilitates STING degradation

The conserved N-terminal Gly residue of proteins is required for N-myristoylation[22]. To clarify the specific target of N-myristoylation that controls STING activation, we first analyzed the amino acid sequences of the important proteins in the cGAS-STING pathway. Cyclic GMP–AMP synthase and STING did not contain the N-terminal Gly residue (Fig. S5a). Adenosine diphosphate-ribosylation factor 1 (ARF1), a master regulator of cargo trafficking through the Golgi apparatus, contains an N-terminal Gly residue (Fig. S5b), and can be N-myristoylated[34]. ARF1 is a core component of the coat protein complex I (COP-I) complex that controls STING membrane trafficking

from the Golgi to the ERGIC, a key step in STING-directed autophagosome formation[8,26]. Myristic acid-induced wild-type ARF1 N-myristoylation in HEK293T cells, but not in the ARF1 G2A mutant (an N-myristoylated-disrupted mutant, in which a Gly to Ala point mutation was introduced) (Fig. 7a). In PMs, N-myristoylated ARF1 was also detected during HSV-1 infection (Fig. 7b). In addition, ARF1 coprecipitated with STING in both HEK293T cells and HSV-1-infected mouse PMs (Fig. 7c, d). These data suggest that myristic acid may target ARF1 to attenuate cGAS-STING activation.

To further confirm the functions of ARF1 in myristic acid-mediated STING degradation, we performed *Arf1* siRNA knockdown experiments in PMs (Fig. S5c). *Arf1* knockdown markedly reversed the inhibitory effects of myristic acid on the protein expression of TBK1 and IRF3 during HSV-1 infection (Fig. 7e). N-myristoylation is crucial for protein shuttling and reversible membrane binding and confers reversible protein association with membranes and other signaling proteins[35]. We then investigated whether myristic acid could regulate the interaction between ARF1 and STING. Myristic acid treatment enhanced the interaction between ARF1 and STING (Fig. 7f). Consequently, ARF1 wild-type, but not ARF1-G2A mutant, suppressed STING protein expression, and myristic acid treatment further enhanced the

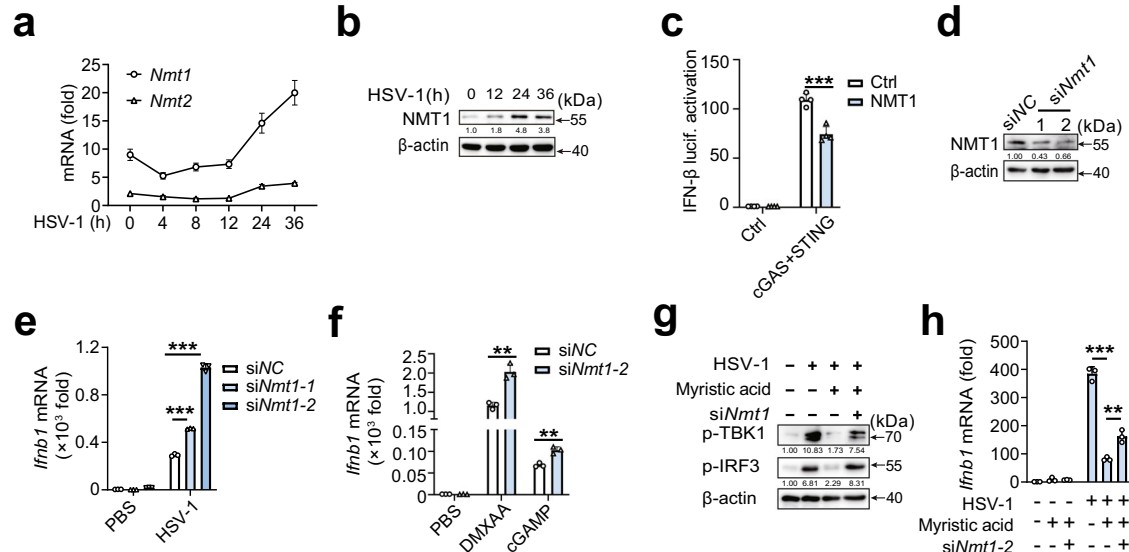

**Fig. 6 | NMT1 attenuates cGAS-STING pathway. a** Quantitative polymerase chain reaction (qPCR) analysis of *Nmt1* or *Nmt2* mRNA in peritoneal macrophages (PMs) during HSV-1 infection, *n* = 3 samples examined over 3 independent experiments. **b** Immunoblot analysis of NMT1 in PMs during HSV-1 infection. **c** Luciferase activity analysis of interferon (IFN)-β activation in HEK293T cells transfected with empty vector or cGAS plus STING plasmids, together with N-myristoyltransferase 1 (NMT1) plasmid or empty vector plasmid, *n* = 4 samples examined over 3 independent experiments. **d** Immunoblot analysis of NMT1 expression in PMs transfected with negative control siRNA (siNC) or *Nmt1* siRNA (si*Nmt1*) for 48 h. **e, f** qPCR analysis of *Ifnb1* mRNA expression in PMs transfected with siNC or si*Nmt1*, followed by HSV-1 infection, DMXAA, or cGAMP stimulation, *n* = 3 samples examined over 3 independent experiments. **g** Immunoblot analysis of p-TBK1 and p-IRF3 in PMs transfected with siNC or si*Nmt1*, and then stimulated with myristic acid, followed by HSV-1 infection. **h** qPCR analysis of *Ifnb1* mRNA in PMs transfected with siNC or si*Nmt1*, and then stimulated with myristic acid, followed by HSV-1 infection, *n* = 3 samples examined over 3 independent experiments. Statistical significance was determined by unpaired two-sided multiple Student's t-tests in (**c**), (**e**), (**f**), and (**h**). Data are shown as mean ± standard deviation (SD) or typical photographs and are representative of three biological independent experiments with similar results. **P < 0.01, ***P < 0.001. Source data is provided in the Source data file.

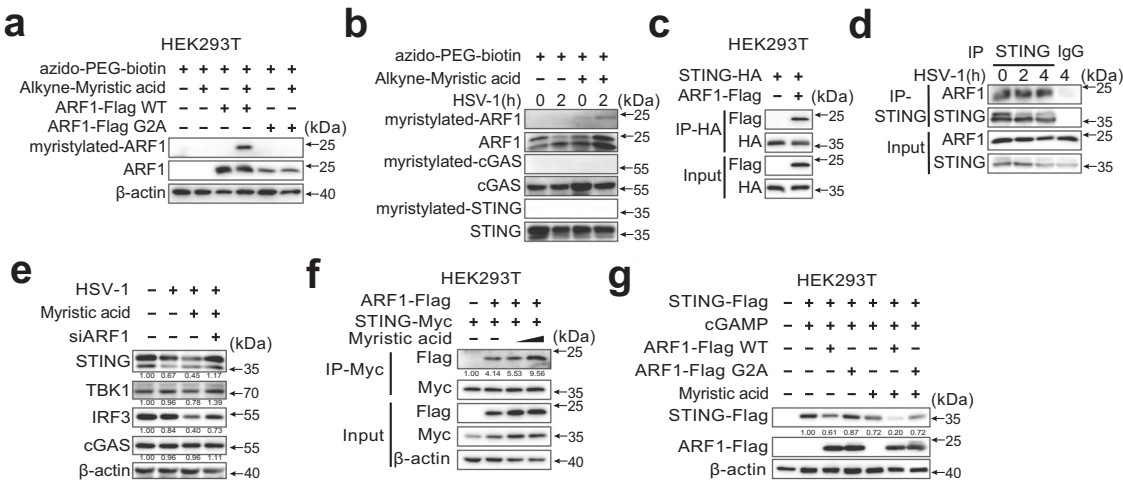

**Fig. 7 | Myristic acid promotes STING degradation by facilitating N-myristoylation of ARF1. a** Immunoblot analysis of N-myristoylation of ARF1 by selective labeling with alkyne-myristic acid in HEK293T cells transfected with wild-type ARF1 or its mutant ARF1 G2A. **b** Immunoblot analysis of N-myristoylation of ARF1, cyclic guanosine monophosphate (GMP)–adenosine monophosphate (AMP) synthase (cGAS), and STING by selective labeling with alkyne-myristic acid in HSV-1 infected peritoneal macrophages (PMs). **c** Co-immunoprecipitation analysis of the interaction between STING-HA with the ARF1-Flag in HEK293T cells. **d** Co-immunoprecipitation analysis of endogenous interaction between ARF1 and STING in herpes simplex virus 1 (HSV-1)-infected PMs. **e** Immunoblot analysis of indicated proteins in PMs transfected with si*NC* or si*Arf1* and then treated with solvent or myristic acid, followed by HSV-1 infection. **f** Co-immunoprecipitation analysis of the interaction between STING-Myc and ARF1-Flag in HEK293T cells, followed by myristic acid treatment. **g** Immunoblot analysis of STING expression in HEK293T cells co-transfected with STING-Flag, plus empty vector, wild type (WT) ARF1 or ARF1 G2A mutant, following myristic acid pretreatment and cGAMP stimulation. Data are shown as typical photographs and are representative of three biological independent experiments with similar results.

inhibitory effects of ARF1 on STING expression (Fig. 7g). Thus, these results indicate that myristic acid enhances N-myristoylation of ARF1 to facilitate STING-dependent autophagy, and therefore enhances the autophagy degradation of the STING complex (see a model in Fig. S6).

## Discussion

STING activation triggers autophagy induction and IFN responses are both highly conserved functions of the cGAS pathway[8]. After cGAMP binds to STING, STING buds from the endoplasmic reticulum into

COP-II vesicles, which then form the ERGIC, where STING recruits TBK1 and IRF3 to initiate IFN responses. The STING-containing ERGIC also serves as a membrane source of LC3 lipidation to induce the formation of autophagosomes that fuse with lysosomes, leading to the degradation of cytosolic DNA or DNA viruses as well as STING proteins[8]. STING-triggered autophagy is vital for the host to eliminate invading pathogens and serves as a self-limiting mechanism of STING-induced IFN responses[8,36]. The balance between STING-dependent autophagy and IFN responses is crucial for the maintenance of immune homeostasis. In this study, we determined that the endogenous SFA myristic acid facilitates STING-dependent autophagy. Myristic acid enhances N-myristoylation of ARF1, a key modification required for its function in directing STING membrane trafficking[34,37], and then limits cGAS-STING-induced IFN responses by facilitating autophagy degradation of STING. Therefore, myristic acid acts as a metabolic checkpoint that balances STING-dependent autophagy and IFN responses.

Perturbation of lipid metabolism is an essential feature of immune cells during viral infections[38–40]. Immune cells undergo alterations in lipid metabolism to enhance antiviral responses and facilitate viral clearance[38]. Cholesterol synthesis is decreased during viral infection, which spontaneously engages in the activation of STING-dependent IFN responses[41]. Upon viral infection, macrophages reduce the expression of 7-dehydrocholesterol reductase (DHCR7, an enzyme that catalyzes 7-dehydrocholesterol [7-DHC] to cholesterol), which causes the accumulation of 7-DHC in macrophages to enhance IRF3-dependent antiviral immunity[42]. Pathogen infection extensively remodels host lipid metabolism to enable the transfer of fatty acids from lipid droplets, which serve as an innate immune hub that is advantageous for organizing an intracellular host defense by accumulating and using immune proteins[40,43]. In contrast, viruses have evolved elaborate strategies to achieve immune escape by exploiting the lipid machinery. For example, viral infection enhances intracellular lipid peroxidation and blocks STING-dependent antiviral responses[17]. Nitro-fatty acids are endogenously formed in response to viral infections. Nitro-fatty acids covalently modify STING by nitro-alkylation, inhibit its palmitoylation, and thus suppress STING-dependent IFN responses and inflammation[44]. Here, we show that myristic acid enhances STING-dependent autophagy, while attenuating IFN responses by facilitating autophagy degradation of the STING complex. Viruses infection downregulates the cellular level of myristic acid to release the self-limitation of STING-dependent IFN responses by suppressing autophagic degradation of STING complex. As a result, the reduction of myristic acid in macrophages caused by viral infection promotes viral clearance by enhancing STING-dependent IFN responses. Therefore, myristic acid may serve as a metabolic checkpoint that determines the consequences of the dynamic interaction between invading viruses and host immune cells.

Aberrant STING activation has been implicated in a variety of autoimmune disorders, inflammatory diseases, metabolic diseases, and cancer[45,46]. Consequently, STING has emerged as an attractive target for pharmacological therapies. For example, nitrofurans (C-176 and C-178) and indole ureas (H-151) both specifically bind to Cys91 of STING and inhibit its palmitoylation and activation, thereby ameliorating STING-dependent immunopathologies both in vitro and in vivo[47]. In the present study, we found that several inhibitors of N-myristoylation, including 2-HOM, DDD85646, and IMP-1088, markedly attenuated STING-dependent IFN responses in macrophages. In addition, myristic acid is a crucial endogenous metabolite that regulates lipid metabolism, which can be supplemented by diet[43]. Recently, dietary palmitic acid, another type of long-chain SFA, has been reported to promote prometastatic memory and potentiate metastasis initiation, suggesting the potential role of dietary SFAs in manipulating cellular status[48]. In the context of HSV-1 infection, myristic acid inhibits STING-dependent IFN responses in vivo. Therefore, myristic acid is speculative to suppress excessive STING activation and thus ameliorate related autoimmune disorders as a dietary supplement.

In summary, our study identified myristic acid as a metabolic checkpoint that balances STING-dependent autophagy and IFN responses and sheds light on the crosstalk between cellular fatty acid metabolism and the DNA-sensing pathway. Furthermore, we suggest myristic acid and N-myristoylation as promising targets for the treatment of diseases caused by aberrant STING activation.

## Methods

### Mice

*Sting*-deficient mice (025805)[49] and *Ifnar1*-deficient mice (028288) were obtained from the Jackson Laboratory. Both male and female C57BL/6J mice (6 to 11 weeks of age) were used in animal experiments that were ethically conducted according to the criteria of the National Institute of Health Guide for the Care and Use of Laboratory Animals and approved by the Scientific Investigation Board of the School of Basic Medical Science, Shandong University, Jinan, Shandong Province, China.

### Reagents

Myristic acid (S5617), stearic acid (S5733), lauric acid (S4726), palmitic acid (S3794), 3-methyladenine (3-MA) (S2767), and chloroquine (S6999) were from Selleck. DDD85646 (GC43386) and IMP-1088 (GC40101) were obtained from GlpBio. 2-HOM (H6771), myristoyl-CoA (870714P), dithiothreitol (DTT, 10197777001), digitonin (D141), ATP (A7699), GTP (V900868), TentaGel™ TBTA (696773), TCEP (C4706), potassium chloride (746436), magnesium chloride (M4880), sucrose (V900116), magnesium chloride (449172), and MG132 (C2211) were purchased from Sigma-Aldrich. Biotin-PEG2-azide (B122224) was purchased from Aladdin. Alkyne-myristic acid (13267) was obtained from Cayman. 2′3-cGAMP (tlrl-nacga23-1), DMXAA (tlrl-dmx), and ISD (tlrl-isdn) were obtained from Invivogen. Biotin-cGAMP (C157) was obtained from BioLog Life Sciences. The sequences of synthetic IFN-stimulating DNA (ISD) were as follows: F 5′-TACAGATCTACTAGTGATCTATGACTGATCTGTACTGATCTACA-3′, R 5′-TGTAGATCATGTA-CAGATCAGTCATAGATCACTAGTAGATCTGTA-3′), which were modified with biotin. HEPES (1939945) was purchased from Gibco. Streptavidin (Sepharose Bead Conjugate) was purchased from Cell Signaling Technology. BSA (10428010110) was obtained from GenView. The concentrations of stimuli or inhibitors used were as follows: myristic acid 200 μM, 2-HOM 100 μM, DDD85646 5 μM, IMP-1088 1 μM, chloroquine 100 μM, 3-MA 250 μM, MG132 10 μM, myristic acid alkyne 50 μM, and myristoyl-CoA 25 μM. Streptavidin (magnetic bead conjugate) (#5947), and anti-rabbit IgG (#2729) were purchased from Cell Signaling Technology. Anti-ICP5 (ab6508), anti-LC3B (ab192890), and anti-GM130 (ab52649) were purchased from Abcam. Alexa Fluor 633 (A-21071) and Alexa Fluor 488 (A-11059) were purchased from Thermo Fisher Scientific. Anti-Myc (M4439) and anti-Flag (F1804) antibodies were purchased from Sigma. Anti-HA (TA180128) was purchased from Origene. All antibodies were diluted with phosphate-buffered saline containing BSA. Protein G agarose (sc-2002) was used for immunoprecipitation, and horseradish peroxidase-conjugated secondary antibodies were obtained from Santa Cruz Biotechnology. Cell Counting Kit-8 (CCK-8) (APE×BIO) was used to quantitatively measure cell viability. The solid myristic acid was dissolved in 0.1 M NaOH solution at 70 °C, and then mixed thoroughly with 10% bovine serum albumin (BSA) (V/V) at 55 °C, and hydrochloric acid was used to adjust the pH to 7.5.

### Stimulants and viruses

Stimulants were used at the following concentrations: ISD, 5 μg/mL; cGAMP, 3 μg/mL; and DMXAA, 150 μg/mL. Sendai virus was purchased from the China Center for Type Culture Collection. Herpes simplex virus type 1 (HSV-1, Kos strain) was a gift from X. Cao (Second Military Medical University, Shanghai, China).

## Cells

To obtain mouse primary peritoneal macrophages (PMs), C57BL/6J mice were administered 3% Brewer's thioglycollate by i.p. injection. Mouse peritoneal exudate cells (PECs) were harvested 3 d later and cultured for another 2 h to remove non-adherent cells. The remaining adherent monolayer cells were used as PMs. Human embryonic kidney (HEK293T) cells, THP-1, and Bj cells were obtained from the American Type Culture Collection. The 293-Dual™ hSTING-A162 cell line stably transfected with the A162 isoform of human STING (S162A) and sensitive to DMXAA, were obtained from InvivoGen. All used cells were authenticated by morphology, karyotyping, and polymerase chain reaction (PCR)-based approaches, and the MycoProbe detection kit (R&D systems) was used to check for mycoplasma contamination, and only the cells without mycoplasma contamination were used. Mouse embryonic fibroblasts (MEFs) were generated from pregnant females for 13- to 14-day-old mice[50]. Briefly, the fetal viscera, head, and limbs were excised from embryos, and the rest of the other embryonic tissues were minced and incubated with 0.25% trypsin-EDTA for 30 min at 37 °C. The MEFs were cultured, and then expanded for subsequent experiments. All the cells were cultured in DMEM supplemented with 10% FCS (Invitrogen-Gibco), 100 U/mL penicillin, and 100 µg/mL streptomycin at 37 °C under 5% $CO_2$.

## Plasmids and transfection

The STING-V147L, STING-N154S, and ARF1-G2A mutants were generated using the KOD-Plus-Mutagenesis kit (Toyobo), and all constructs were confirmed by DNA sequencing. The IFN-β reporter plasmid, expression plasmids for cGAS, STING, TBK1, and IRF3 5D were described before[51]. The plasmids of NMT1-Myc were procured from Origene, and the plasmids of ARF1-Flag were obtained from Sino Biologicals. Lipofectamine 2000 reagent (Invitrogen) was used for transfection in MEFs and HEK293T cells.

## RNA interference

The target sequences for transient silencing are as following: 5-GGA AACUGGUUGGGUUCAUUUAUGAACCCAACCAGUUUCCTT-3 (siRNA1) and 5-GGAAGCUGAUUGAAGUGAAUUUCACUUCAAUCAGCUUCCTT-3 (siRNA 2) for mouse *Nmt1*; 5-GGGAAGACAACAAUUCUAUUUAUA GAAUUGUUGUCUUCCCTT-3 (siRNA1), 5-GAGGAUGCUAGCUAGCUGA AGAUUUAUCUUCAGCUAGCAUCCUCTT-3 (siRNA 2) and 5-GCUCUAU GAAGGACUAGAUUUAUCUAGUCCUUCAUAGAGCTT-3 (siRNA3) for mouse *Arf1*; 5-UUCUCCGAACGUGUCACGU-3 for mouse 'negative control.' siRNA duplexes were transfected into primary peritoneal macrophages using the INTERFERin® Reagent (Polyplus-transfection).

## Viral infection in vivo

C57BL/6J mice (8-week-old females) were administered intraperitoneally (i.p.) with myristic acid (100 mg/kg) 24 h before herpes simplex virus 1 (HSV-1, Kos strain) infection by i.p. ($2 \times 10^7$ p.f.u. per mouse). C57BL/6 J mice (8-week-old females) were orally administered with DDD85646 (50 mg/kg) twice a day for three days before HSV-1 infection by i.p. ($2 \times 10^7$ p.f.u. per mouse). The serum of the mice was collected for enzyme-linked immunosorbent assay (ELISA) analysis of interferon (IFN)-β at 8 h after HSV-1 infection. Lungs from phosphate-buffered saline (PBS)- or HSV-1-injected mice were collected for immunohistochemical staining with ICP5 or hematoxylin-eosin staining and examined by light microscopy for HSV-1 replication or histological changes. Mice were euthanized 72 h after HSV-1 infection ($2 \times 10^7$ p.f.u. per mouse) to obtain brain, spleen, and lung tissues for quantitative PCR (qPCR). For virus infection survival experiments, mice were treated with myristic acid 24 h before HSV-1 infection, and the mice were monitored for survival after viral infection.

## Luciferase activity assay

Luciferase activity was measured using the Dual-Luciferase Reporter Assay System (Promega), as previously described[51]. Data were normalized for transfection efficiency by dividing the activity of firefly luciferase with that of Renilla luciferase. For HEK293T cells, a mixture of the indicated luciferase reporter plasmid, pRL-TK-Renilla-luciferase plasmid, and the indicated expression plasmids were co-transfected into cells using Jet-PEI transfection reagent (Polyplus). The solvent or myristic acid (200 µM) was added to each sample per well of a flat-bottom 96-well plate after 4 h of transfection. Luciferase activity was measured in accordance with the manufacturer's instructions after 24 h. For 293T-Dual™ hSTING-A162 cells, 2-HOM (100 µM) or DDD85646 (5 µM) were added to each well, and then DMXAA was added to each well. Pipet 10–20 µL of 293T-Dual™ hSTING-A162 cell culture supernatant per well in a 96-well white plate. Subsequently, 50 µL of QUANTI-Luc™ was added per well, and luciferase activity was measured.

## ELISA

Interferon-β in the cellular supernatant or serum was quantified using ELISA kits (BioLegend). Intracellular cGAMP levels were measured using ELISA kits (Cayman) according to the manufacturer's instructions.

## Real-time-PCR

Total RNA was extracted using the RNAfast200 RNA extraction kit (Fastagen). RNA was reverse-transcribed into cDNA using reverse transcriptase (Vazyme), and quantitative real-time PCR analysis was performed using SYBR Green RT-PCR kits (Vazyme). Data were normalized to *β-actin* expression in each sample. The primers used for real-time PCR assays are listed in Supplemental Data Table 1.

## Immunoblot analysis

Cells were lysed using RIPA reagent (Pierce, Thermo Scientific) supplemented with a protease inhibitor cocktail (Sigma-Aldrich). Protein concentrations were measured using the Pierce BCA Protein Assay Kit (Thermo Scientific), and the cell lysates were denatured with loading buffer. For immunoblot analysis, immunoprecipitates or whole-cell lysates were loaded and separated by sodium dodecyl sulfate (SDS)-polyacrylamide gel electrophoresis (PAGE), and the proteins were transferred onto nitrocellulose membranes (Millipore) for immunoblot analysis as described previously[51]. The following antibodies were used for immunoblot analysis: Anti-STING (1:1000, #13647), anti-TBK1 (1:1000, #3504), anti-p-IRF3 (1:1000, #4947), anti-IRF3 (1:1000, #4302), anti-p-STAT1 (1:1000, #9167), anti-cGAS (1:1000, #31659), and anti-streptavidin-HRP (1:1000, #3999) were purchased from Cell Signaling Technology. Anti-NMT1 (1:2000, ab186123), anti-ICP5 (1:3000, ab6508), anti-LC3B (1:2000, ab192890), and anti-p-TBK1 (1:1000, ab109272) were purchased from Abcam. Anti-Myc (1:5000, M4439) and anti-Flag (1:5000, F1804) antibodies were purchased from Sigma. Anti-β-actin (1:20,000, 66009-I-Ig) and anti-ARF1(1:1000, 20226-1-AP) were purchased from Proteintech. Anti-HA (1:2000, TA180128) was purchased from Origene.

## Immunoprecipitation

HEK293T cells and mouse PMs were lysed with IP lysis buffer, containing 50 mM Tris-HCl (pH 7.4), 50 mM EDTA, 150 mM NaCl, and 1% NP-40 and mixed with a protease inhibitor cocktail (Sigma-Aldrich). The supernatants of cell lysates after centrifugation were collected and immunoprecipitated with the indicated antibodies with Protein A/G PLUS-Agarose (Santa Cruz) overnight at 4 °C. The beads were washed five times with IP buffer, and the immunoprecipitates were eluted by boiling in IP buffer containing 1% (w/v) SDS.

## Immunofluorescence and confocal microscopy analyses

The MEFs or Bj cells were plated onto glass coverslips in 24-well plates. The cells were fixed with Immunol Staining Fix Solution (Beyotime), permeabilized with 0.5% Triton-X 100 in PBS, and blocked in 3% BSA for 1 h. The STING, LC3 or GM130 were labeled with primary antibodies and stained with a secondary antibody conjugated to either Alexa Fluor 633 or Alexa Fluor 488. Nuclei were stained with DAPI for 3 min (Beyotime). The cells were subjected to microscopy analysis with a Zeiss LSM880 confocal laser microscope, which was provided by the Micro Characterization Facility of Shandong University.

## Validation of protein N-myristoylation using click chemistry

HEK293T cells or macrophages were incubated with 50 μM myristic acid alkyne (Cayman) for 24 h. Cells were lysed on ice with PBS containing 0.4% SDS and protease inhibitors, and then the lysates were incubated with 1 mM $CuSO_4$, 100 μM TBTA ligands, 100 μM Biotin-PEG2-azide (Aladdin), and 1 mM TCEP for 1 h at 25 °C. After precipitation with cold acetone, the protein precipitates were dissolved in IP buffer containing 0.1% SDS and immunoprecipitated with streptavidin beads for 3 h at 4 °C. The loading buffer was used to elute proteins at 95 °C after washing the beads six times, and then SDS-PAGE was used to separate the proteins.

## Fatty acid quantification by gas chromatography (GC)–mass spectrometry (MS)

The cell samples were added with 1 mL extract of chloroform: methanol (2:1 v/v) and vortexed for 1 min. Then, 0.5 mL of phosphate-buffered saline (PBS) was added to the samples and mixed thoroughly to remove the supernatant. Nitrogen was used to dry the samples, mixed with 2 mL of acidified methanol, and incubated at 80 °C for 2 h. After methyl esterification, the samples were extracted with hexane (1 mL of hexane). To quantify medium- and long-chain fatty acids, Supelco 37-component FAME (fatty acid methyl ester) mix (Sigma-Aldrich) was used to construct a calibration curve for the concentration range of 0.5–1000 mg/L. The extracted sample was analyzed by GC–MS operating in a single ion monitoring mode using an Agilent 7890B-7000D system with an Agilent DB-WAX capillary GC column. The initial temperature was 50 °C, maintained at this temperature for 3 min, increased to 220 °C at 10 °C/min, and maintained at 220 °C for 20 min. A quality control sample was used to test and evaluate the stability and repeatability of the system. The temperature was set to 280 °C for the injection port and 230 °C for the ion source. The electron bombardment ionization source, selected ion monitor scanning mode, and electron energy were 70 eV. Mass data were analyzed using ChemStation (Agilent) to determine the concentration of each compound.

## Statistics and reproducibility

Data from three experiments are presented as the mean ± standard deviation (SD). Statistical analyses were performed using GraphPad Prism 8. All quantitative measurements were tested with a normal distribution. Comparisons between two groups were performed using the unpaired two-tailed Student's t-test, with a $P$-value < 0.05, considered statistically significant. Survival curves were compared using the Kaplan–Meier survival method. No statistical method was used to predetermine sample size and no data were excluded from the analyses.

## Reporting summary

Further information on research design is available in the Nature Portfolio Reporting Summary linked to this article.

## Data availability

All data supporting the findings of this study are available within the article and the data generated in this study are provided in the Supplementary Information and Source data file. Source data are provided with this paper.

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

## Acknowledgements

This work was supported by grants from the National Natural Science Foundation of China (grant nos. 82125020 and 31870866 to W.Z., grant no. 82101855 to M.J.) and the Postdoctoral Science Foundation of China (grant no. 2021T140406 to M.J.).

## Author contributions

W.Z. and M.J. conceived the project, designed the experiments, analyzed the data, and wrote the manuscript. M.J., Y.W., and J.W. performed most of the experiments. D.Q., M.W., L.C., Y. F., and C.Z. assisted with the experiments and provided technical help. C.G. and J.J. provided expertise and advice. W.Z. provided an overall direction.

## Competing interests

The authors declare no competing interests.
