## [Peer Review File · Nature Communications]

Myristic acid as a checkpoint to regulate STING-dependent autophagy and interferon responses by promoting N-myristoylationReviewers' comments:

Reviewer #1 (Remarks to the Author):

In this work Jin et al suggest Myristic acid to be a key regulator of IFN signaling through the STING pathway. The authors suggest that myristoylation of ARF1 drives STING trafficking to the autophagy pathway and not the IFN pathway. This is an interesting idea, but is not supported by the data as they stand now, and I find the work underdeveloped for consideration for publication in Nature Communications.

1. Figure 2. The authors need to provide data on virus titer and inflammatory cytokines
2. The data in Fig 2, could also be due to a direct antiviral activity of Myristic acid independent of STING. The authors should test whether Myristic acid affect HSV1 replication in HSV1 permissive STING deficient cells
3. Figure 4j. Macrophages are not a physiologically relevant cell type to evaluate HSV1 replication. Besides, the statement that early clearance of virus is mediated by autophagy is not supported by data. There are other possible interpretations of these data, the most obvious one would be direct antiviral activity of myristic acid
4. Figure 5. The inhibitor-based data alone, cannot support the conclusion "Myristic acid inhibits cGAS-STING activation via facilitating N-myristoylation"
5. Figure 5. The authors do not provide strong data on the role of Myristic acid inhibitors and cGAS-STING-activated autophagy.
6. Most data are based on exogenous addition of Myristic acid. There is a critical lack of data on the endogenous role of Myristic acid in regulation of the pathway. The work would gain substantially by demonstrating an in vivo phenotype of Mnt1^{-/-} mice e.g. during HSV-1 infection
7. The proposed mechanism on myristoylation and STING trafficking is not supported by mechanistic data. For instance, there are no data showing enhanced loading of STING into COP-II vesicles after myristoylation or STING ER-to-Golgi trafficking.
8. As a general – but very important – comment, the claim in the title that the manuscript explains how myristoylation governs the balance between IFN and autophagy responses are not found in the data. Besides, if STING autophagy was independent of ER exit it would be inconsistent with current models (e.g. Nature volume 567, pages262–266 (2019)). It does not explode that it may be correct, but it would require some mechanistic explanation, and several controls.

Reviewer #2 (Remarks to the Author):

Myristic acid acts as a metabolic checkpoint that balances STING-dependent autophagy and IFN responses

In this paper, Jia & al. report that the fatty acid Myristic acid attenuates cGAS-dependent signaling and therefore it is an endogenous suppressor of cGAS-dependent antiviral innate immunity by specifically inhibiting the palmitoylated STING protein. Additionally, they claim that myristic acid promotes STING complex autophagy degradation by enhancing overall cellular N-myristoylation. However, they propose that STING degradation mediated by myristic acid is specifically due to enhance ARF1 N-myristoylation. This is an interesting well-written paper. At the present, however, there are several issues that should be addressed to back up their model and the conclusions made, as outlined below.

Comments:

1) My main concern is that the conclusions held in the text are not always supported by the figures and the appreciation of this work has also been reduced by the incorrectly quoted figures and the sometimes erratic order of the figures. In all these cases, I will make a list below.

Line 86: Fig. 1b is incorrect. The authors refer here to Fig. 1c.

Line 92: The authors claim that Myristic Acid decreased HSV-1 infection, and ISD induced the expression of Tnfa and Il6. really don't see what they say in Fig. 1d at all.

Line 117-124: all of Fig. 3 is unclear. There is discrepancy between the panels in the Fig. and what it is written in the results.

Line 127: I do not see the dose-dependent inhibition of the STING protein in Fig. 4a, as claimed by the authors. The figure needs to be increased in size for better visibility and quantification of bands are required. Moreover, it is missed the control + Myristic Acid –HSV-1.

Line 129: I disagree with the statement about irf3 (Fig. S2). There is an effect at 2h for this protein.

Line 153-154: Fig. S3a is incorrect. The authors refer here to Fig. S3b. Additionally, I disagree with authors' conclusions. I do not see any increase in Myristoylated proteins taking also in account B-actin level. Authors must provide MS analysis of these samples to support their hypothesis. I'm also confused about the conclusion from the immunoblot analysis of the Fig. I do not see any phosphorylation induction by Myristoy-CoA. Conversely, I can see an inhibition for IRF3. Again, quantification of the replicates is required.

Line 176: Nmt1 knockdown partially reverses the decrease in phosphorylation of TBK1 and mRNA Irfb1. This should be correct.

2) The Legends of Figures are very often imprecise, a lot of information is missing and what it is reported in the legend is not in accordance with what is written in the results section. For instance: Line 91: in Fig. 1b it is reported that it is an analysis of IFN-B secretion. However, in the result text it is reported it is IFN-B expression. Only Fig. 1c reports mRNA expression level. Authors need to be more careful in their explanations in both text and legends.

Lane 99: There is no information found about the immunoblot assays, particularly the antibodies used.

3) The authors ping-pong between experiments in PMs with endogenous proteins and analysis in HEK293T cells transiently transfected with different plasmids without providing details nowhere including M&M, once again making it difficult to appreciate the appropriateness of the results' conclusions. The authors should explain why these types of experiments in HEK293T cell with overexpressed proteins instead to make the experiments in MPs and looking at the endogenous proteins.

This is the case, for example, of Fig. 3a, whose insufficient legend and shown results do not help to understand the authors' conclusion.

In some cases, as in Fig 4c, I do not know if experiments have been done in HEK cells or MPs. The authors need to better clarify all these points.

4) Quantification of all immunoblot bands should be made and reported together with further information on Western blots, including the antibodies used.

5) Lane 173: increase of NMT1 expression should be revealed by immunoblot using Ab against NMT1. The authors have this Ab.

6) Line 189: why 2 bands for Arf1 in Fig. 7b but not in all other panels?

7) What can we say about the ratio between Myristoylated ARF1 and the total amount of ARF1?

8) Line 132: where are the experiments with MG132?

9) Fig. 4c: why does STING migrate differently depending on the conditions? And more particularly why detection in Fig. 4c is not similar to that in Fig 4a and Fig. 4c and 4d –chloroquine and + Myristic acid are not equivalent.....?

10) Line 142: specify that it is after cGAMP and HSV-1 stimulation.

11) A current model of how TBK1 and IRF3 start IFN1 expression would have helped to better understand what it is known in the literature and what the authors bring as new knowledge.

12) Finally, I have to confess that the nomenclature can be hard to follow and slowed my appreciation of this paper. There isn't a list of abbreviations and it was hard to search where the full name was cited at the beginning.

Reviewer #3 (Remarks to the Author):

In this well-written manuscript, the authors provide intriguing evidence that the long-chain fatty acid, myristic acid, might act as a metabolic checkpoint that regulates the balance between STING-dependent type I interferon responses and the ability of this innate immune signaling component to induce autophagy in elicited murine macrophages. The present studies are interesting and provide insights into the mechanisms by which potentially detrimental immune responses to viral challenge are regulated. The mechanistic investigations described are generally thorough and convincing, but the present study suffers from limitations related to the selection of elicited peritoneal macrophages for study and inadequately addressed virological aspects.

MAJOR POINTS

1. The reason for the selection of elicited peritoneal macrophages in these studies is unclear since other immune cell types such as dendritic cells are more important sources of type I interferon production. Also, no attempt is made to verify the present findings in bone marrow-derived macrophages or human myeloid cells.
2. It is stated that no other short chain fatty acids have an effect, but c16:1 looks close to being so in Figure 1A but fails to be significant perhaps due to the limited sample size. Similarly, palmitic acid and linoleic acid show 15-25% reductions in HSV-1 infection-induced *Irfn1* mRNA expression in Figures 1B and S1A. It should also be noted that no other SFAs were studied further and so the use of the term “selectively” cannot be used.
2. In contrast to the authors’ description in line 99, SeV does elicit a decrease in pTBK1 levels in the blot shown in Figure 1f. Similarly, myristic acid is stated to have no effect on mRNA expression in Figure S2 but significant differences are shown in STING and *Irf3* mRNA levels.
3. Figure 1A shows considerable variability suggesting an insufficient sample size and the p-TBK1 blot is of poor quality in Figure S1c. The myristic acid-induced decreases in p-IRF3 in Figs 3F and G, and p-TBK1 in Fig. 3F, are not convincing. Also, please provide quantification of protein expression for Figures 5i and S4c, and quantification is lacking for the effects in Figure 7e.
4. It is difficult to compare findings between stimuli and readouts, as readout times either differ between stimuli or are not described. In addition, the numbers of experimental repeats are not provided in the figure legends for all experiments.
5. In Figure 2, the descriptions of the *in vivo* treatments and HSV-challenge are not sufficient, and the viral strains used in these studies and their validation are not described. The study of the effect of treatments on HSV are limited to assessments of UL30 mRNA expression. It is unclear why HSV burden and leukocyte recruitment was limited to lung tissue.
6. The authors have a tendency to overstate their results, e.g. on lines 164 and 176, effects are described as be “reversed” but are only partially so. Similarly, the approximately 30% inhibition by NMT1 is described on line 174 as being “markedly”. In lines 235-242, please indicate that this is hypothesized from the present study. Also, the notion that myristic acid could ameliorate autoimmune disorders (Lines 254-256) is highly speculative and is not supported by the present study. Finally, the descriptive title of Figure S1 is inaccurate as the data provided does not definitively show inhibition of the cGAS pathway, and in line 115, cGAS dependence was not proved.

MINOR POINTS

1. The abstract does not denote the species and cell types studied.
2. Figure 1C is incorrectly referenced as Figure 1B on line 86.
3. In Figure 4B, protein expression is labeled as Myc when STING appears to be intended.
4. In line 228, please expound on why such an immune hub is “advantageous”.
5. Please address the following grammatical/syntactical errors:
Line 92 change to “infection- and ISD-induced expression”
Line 99 change to “but not SeV, induced phosphorylation”
Line 206 change to “triggered autophagy induction and IFN production are”
Line 211 change to “proteins”
Line 215 change to “we determined that the endogenous SFA myristic acid facilitates”
Line 231 change to “Nitro-fatty acids are endogenously formed in”
Line 671, change “extended” to “supplemental”

Reviewer #1:

In this work Jia et al suggest Myristic acid to be a key regulator of IFN signaling through the STING pathway. The authors suggest that myristoylation of ARF1 drives STING trafficking to the autophagy pathway and not the IFN pathway. This is an interesting idea, but is not supported by the data as they stand now, and I find the work underdeveloped for consideration for publication in Nature Communications.

Answer: We appreciate your precious time in reviewing our manuscript. In accordance with your valuable suggestions and comments, we carefully revised the manuscript, and several additional experiments were performed. In doing so, we have strengthened mechanistic details and the physiological relevance of our findings.

1. Figure 2. The authors need to provide data on virus titer and inflammatory cytokines

Answer: Thanks for the valuable suggestions and we performed additional experiments accordingly. We further examined HSV-1 virus replication in different tissues by detecting HSV-1 UL30 mRNA expression, a method that was widely used to examine viral replication (*Nat Immunol.* 2020; 21:727-735.; *Nat. Commun.* 2017; 8:15534.; *Cell Mol. Immunol.* 2018; 15:907-916.).

The replication of HSV-1 in the lung, brain, and spleen was enhanced in myristic acid-treated mice than in control mice at 72 hours after viral infection as monitored by the expression of the HSV-1 UL30 mRNA (Fig. 2e). Furthermore, myristic acid-treated mice produced less IFN- β , IL-6, and TNF- α than that of control mice during HSV 1 infection (Fig. 2b). We added these new data in the revised manuscript.

2. The data in Fig 2, could also be due to a direct antiviral activity of Myristic acid independent of STING. The authors should test whether Myristic acid affect HSV1 replication in HSV1 permissive STING deficient cells

Answer: We accepted the valuable suggestion and performed the suggested experiments. Myristic acid treatment enhanced HSV-1 replication both in wild-type PMs and mouse embryonic fibroblasts (MEFs), but in STING-deficient PMs and MEFs (Fig. 4i and 4j). These results indicated that the function of myristic acid in HSV-1 infection was dependent on STING, and we added these new data in the revised manuscript.

3. Figure 4j. Macrophages are not a physiologically relevant cell type to evaluate HSV1 replication. Besides, the statement that early clearance of virus is mediated by autophagy is not supported by data. There are other possible interpretations of these data, the most obvious one would be direct antiviral activity of myristic acid.

Answer: Yes, this is a good question. Macrophages are at the front line of defense against viruses and constitute a major part of the innate immune systems. Upon virus infection, pattern recognition receptors (PRRs) from macrophages are activated and produce IFN α/β , which induces the expression of several hundred ISGs, a large number of which function to

induce an antiviral state within the cell and restrict the viral replication cycle in cells that have already been infected (*Trends Microbiol.* 2013;21(8):380-8; *Nat. Rev. Immunol.* 2015;15(2):87-103.). Thus, many studies have investigated viral replication in macrophages to illustrate the effect of IFN α/β on viral replication. (*Immunity.* 2020;53(1):115-126; *Immunity.* 2014;40(3):329-41; *Science.* 2019;365(6454): eaav0758.). Furthermore, several studies also examined the effect of STING activation on HSV-1 replication in MEFs (*Nat. Immunol.* 2019 Feb;20(2):152-162; *Immunity.* 2014 Mar 20;40(3):329-41), so we further performed the experiments in MEFs. Myristic acid treatment enhanced HSV-1 replication in wild-type MEFs, but not in *Sting*-deficient MEFs (Fig. 4j).

Next, we investigated whether myristic acid has direct antiviral activity during HSV-1 infection. Myristic acid treatment greatly inhibited HSV-1 replication after 4 hours of infection in wild-type PMs (approximate 4 times lower), and STING deficiency could not completely block the inhibitory effect of myristic acid treatment in HSV-1 replication (approximate 0.25 times lower) (A), suggesting an inhibitory effect of myristic acid partially due to its direct antiviral activity. It is difficult to distinguish how much of the function of myristic acid in early HSV-1 infection (4 hours) is attributable to STING-mediated autophagy or its antiviral activity. However, STING-deficient completely blocked the enhancement effects of myristic acid treatment in HSV-1 replication after 12 hours of infection (A), which attributed to the impairment of STING-mediated IFN- β signaling. Because of this uncertain mechanism in HSV-1 early infection, we deleted the results of HSV-1 infection before 4 hours and mainly discussed the effect of myristic acid on IFN- β responses in HSV-1

replication at later times.

4. Figure 5. The inhibitor-based data alone, cannot support the conclusion “Myristic acid inhibits cGAS-STING activation via facilitating N-myristoylation”

Answer: We performed the experiment not only with inhibitors but also overexpression N-myristoyltransferase 1 (NMT1) in HEK293T cells (Fig. 6c) and knockdown of NMT1 in PMs (Fig. 6d–6h). The results showed that NMT1 overexpression inhibited cGAS-STING-triggered IFN-β reporter activation. *Nmt1* knockdown substantially promoted HSV-1 infection-, cGAMP-, and DMXAA-induced *Ifnb1* expression in PMs (Fig. 6d–6f). Moreover, *Nmt1* knockdown partially reversed the decrease in phosphorylation of TBK1 and IRF3 and *Ifnb1* mRNA expression in myristic acid-treated PMs (Fig. 6g and 6h). These results indicate that myristic acid inhibits STING activation by regulating N-myristoylation.

5. Figure 5. The authors do not provide strong data on the role of Myristic acid inhibitors and cGAS-STING-activated autophagy.

Answer: Thanks for the valuable suggestions. NMT inhibitor DDD85646 treatment inhibited cGAMP-induced LC3-I to LC3-II conversion in wild-type PMs, but not in STING-deficient PMs (Fig. 5j), suggesting NMT-induced N-myristoylation inhibited STING-mediated autophagy. We added these new data in the revised manuscript.

Figure 5j

6. Most data are based on exogenous addition of Myristic acid. There is a critical lack of data on the endogenous role of Myristic acid in regulation of the pathway. The work would gain substantially by demonstrating an *in vivo* phenotype of *Mnt1*^{-/-} mice e.g. during HSV-1 infection

Answer: Yes, this is a great question. However, because NMT1 is essential for the early development of mouse embryos and proper monocytic differentiation of the mouse bone marrow cells, *Nmt1*-deficient mouse embryonic stem cells resulted in the drastic reduction of macrophages and cause embryonic lethality (*J Immunol.* 2008;180(2):1019-28.). It has been reported that oral administration of DDD85646 (NMTs inhibitor) could reduce the enzyme activity of NMTs *in vivo* (*Nature.* 2010;464(7289):728-32.). To confirm the endogenous role of myristic acid in the regulation of STING pathway, we constructed the animal model of DDD85646 oral administration. DDD85646-treated mice produced higher IFN-β, IL-6, and TNF-α than that of control mice during HSV-1-infection (Fig. 5k). The replication of HSV-1 in the lung, brain, and spleen was reduced in DDD85646-treated mice than in control mice (Fig. 5l). These results suggested that the endogenous myristic acid inhibited cGAS-STING-dependent antiviral innate immunity. We added these new data in the revised manuscript.

Figure 5k

Figure 5l

7. The proposed mechanism on myristoylation and STING trafficking is not supported by mechanistic data. For instance, there are no data showing enhanced loading of STING into COP-II vesicles after myristoylation or STING ER-to-Golgi trafficking.

Answer: We accepted the valuable suggestion and performed the suggested experiments. Upon STING activation, a part of STING traffic from the ERGIC to the Golgi, where STING activates TBK1 and leads to induction of type I IFN, but other parts of STING at the ERGIC lead to the formation of autophagosomes (*Nature*. 2019 Mar;567(7747):262-266.). B₂ cells have previously been used to detect the trafficking of STING (*Nature*. 2019 Mar;567(7747):262-266.). Myristic acid treatment suppressed the cGAMP-induced localization of STING at the Golgi (Fig. 4f), suggesting myristic acid facilitates STING-dependent IFN responses. We added these new data in the revised manuscript.

Figure 4f

8. As a general – but very important – comment, the claim in the title that the manuscript explains how myristoylation governs the balance between IFN and autophagy responses are not found in the data.

Answer: Thanks for the valuable suggestions. Autophagy induction and IFN responses triggered by STING activation are both highly conserved functions of the cGAS pathway. The balance between STING-dependent autophagy and IFN responses is vital for eliminating invading pathogens and maintaining immune homeostasis. Our results suggest that myristic acid limits cGAS-STING-induced IFN responses by facilitating STING-dependent autophagy. Therefore, myristic acid plays a vital role in maintaining the balance of STING-dependent autophagy and IFN responses. To better describe the findings, we have changed the title to “Myristic acid acts as a metabolic checkpoint that regulates STING-dependent autophagy and IFN responses by promoting N-myristoylation of ARF1”.

9. Besides, if STING autophagy was independent of ER exit it would be inconsistent with current models (e.g. Nature volume 567, pages262–266 (2019)). It does not explode that it may be correct, but it would require some mechanistic explanation, and several controls.

Answer: Thanks for the valuable suggestions. Upon cGAMP binding, STING translocation

from the endoplasmic reticulum to the ERGIC and Golgi, where is required for maximal signal transduction of STING-dependent IFN- β responses. Some STING-coated ERGIC vesicles serve as the membrane source for modification by the ubiquitin-like protein LC3 and lead to the formation of autophagosomes, while other STING-coated ERGIC vesicles continue to traffic to Golgi. ARF1-mediate COP-I vesicles are a vital component for STING trafficking, which promotes retrograde STING traffic from Golgi to ERGIC, our results suggest that myristic acid promotes N-myristoylation of ARF1 enhanced STING-dependent autophagy induction, resulting in limiting of STING-dependent IFN- β production.

Reviewer #2:

In this paper, Jia & al. report that the fatty acid Myristic acid attenuates cGAS-dependent signaling and therefore it is an endogenous suppressor of cGAS-dependent antiviral innate immunity by specifically inhibiting the palmitoylated STING protein. Additionally, they claim that myristic acid promotes STING complex autophagy degradation by enhancing overall cellular N-myristoylation. However, they propose that STING degradation mediated by myristic acid is specifically due to enhance ARF1 N-myristoylation. This is an interesting well-written paper. At the present, however, there are several issues that should be addressed to back up their model and the conclusions made, as outlined below.

Answer: We appreciate your precious time in reviewing our manuscript. In accordance with your valuable suggestions and comments, we carefully revised the manuscript, and several additional experiments were performed. In doing so, we have strengthened mechanistic details and the physiological relevance of our findings.

Comments:

1) My main concern is that the conclusions held in the text are not always supported by the figures and the appreciation of this work has also been reduced by the incorrectly quoted figures and the sometimes erratic order of the figures. In all these cases, I will make a list below.

Answer: Thank you very much for your valuable suggestions and we are very sorry for these careless mistakes.

Line 86: Fig. 1b is incorrect. The authors refer here to Fig. 1c.

Answer: We have corrected it in the revised manuscript.

Line 92: The authors claim that Myristic Acid decreased HSV-1 infection, and ISD induced the expression of *Tnfa* and *Il6*. really don't see what they say in Fig. 1d at all.

Answer: We have corrected it as “Myristic acid treatment decreased HSV-1-infection- and ISD-induced the expression of tumor necrosis factor- α (*Tnfa*), and interleukin-6 (*Il6*).”.

Line 117-124: all of Fig. 3 is unclear. There is discrepancy between the panels in the Fig. and what it is written in the results.

Answer: We have rewritten this paragraph as below:

“To clarify the mechanism by which myristic acid attenuates cGAS-dependent signaling, we first performed luciferase reporter assays. cGAS–STING-induced IFN- β luciferase activation was considerably inhibited in myristic acid-treated HEK293T cells, whereas no differences in TBK1- and IRF3-induced IFN- β luciferase activation were observed (Fig. 3a), suggesting that myristic acid may target cGAS or STING. However, myristic acid had no effect on the binding activity of cGAS with ISD (Fig. 3b). Myristic acid also did not decrease cGAMP production during HSV-1 infection (Fig. 3c). These results suggested that myristic acid may target the downstream of STING activation. Next, we further detected the function of myristic acid in cGAMP and STING agonist 5,6-dimethylxanthene-4-acetic acid (DMXAA) induced STING activation. Myristic acid treatment markedly inhibited cGAMP- and DMXAA-induced IFN- β secretion and mRNA expression (Fig. 3d and 3e), and phosphorylation of TBK1, IRF3, and STAT1 (Fig. 3f and 3g). Collectively, these data indicate that myristic acid targets STING, but not cGAS.”

Line 127: I do not see the dose-dependent inhibition of the STING protein in Fig. 4a, as claimed by the authors. The figure needs to be increased in size for better visibility and quantification of bands are required. Moreover, it is missed the control + Myristic Acid – HSV-1.

Answer: Thanks for the valuable suggestions, and we re-performed the experiment in 4a and added the group of Myristic Acid treatment without HSV-1 infection. The results showed that myristic acid treatment inhibited STING, TBK1, and IRF3 protein expression in a

dose-dependent manner (Fig. 4a). We added this new data in the revised manuscript.

Figure 4a

Line 129: I disagree with the statement about irf3 (Fig. S2). There is an effect at 2h for this protein.

Answer: We accepted the valuable suggestion. These results may cause by the limited sample size, so we re-performed the experiment and increased the sample size. The results showed that myristic acid treatment did not affect the mRNA expression of *Irf3*, *Tbk1*, and *Sting* in HSV-1-infected PMs (Fig. S2).

Figure S2

Line 153-154: Fig. S3a is incorrect. The authors refer here to Fig. S3b.

Answer: We have corrected it in the revised manuscript.

Additionally, I disagree with authors' conclusions. I do not see any increase in Myristoylated proteins taking also in account B-actin level. Authors must provide MS analysis of these samples to support their hypothesis.

Answer: Thanks for the valuable suggestions, and we added the analysis of grey value in the revised manuscript. Myristic acid treatment induced high levels of protein N-myristoylation

in PMs (lane 1 compares with lane 3, and lane 2 compares with lane 4), and DDD85646 as inhibitors of myristoylation inhibited myristic acid treatment induced protein N-myristoylation (lane 3 compares with lane 5, and lane 4 compares with lane 6) (Fig. S3a), suggesting that intracellular protein N-myristoylation can be induced by exogenous myristic acid.

Figure S3a

I'm also confused about the conclusion from the immunoblot analysis of the Fig. I do not see any phosphorylation induction by Myristoyl-CoA. Conversely, I can see an inhibition for IRF3. Again, quantification of the replicates is required.

Answer: Thanks for the valuable suggestions, and we re-performed the experiment with the analysis of grey value in the revised manuscript. Myristoyl-CoA treatment enhances HSV-1 induced phosphorylation of TBK1 and IRF3, suggesting a regulatory role for N-myristoylation in the cGAS-STING pathway (Fig. S3b).

Figure S3b

Line 176: Nmt1 knockdown partially reverses the decrease in phosphorylation of TBK1 and mRNA Ifnb1. This should be correct.

Answer: Thanks for the valuable suggestions, and we have corrected it as “*Nmt1* knockdown partially reversed the decrease in phosphorylation of TBK1 and IRF3 and *Ifnb1* mRNA expression in myristic acid-treated PMs (Fig. 6g and 6h).”

2) The Legends of Figures are very often imprecise, a lot of information is missing and what it is reported in the legend is not in accordance with what is written in the results section. For instance:

Line 91: in Fig. 1b it is reported that it is an analysis of IFN-B secretion. However, in the result text it is reported it is IFN-B expression. Only Fig. 1c reports mRNA expression level. Authors need to be more careful in their explanations in both text and legends.

Answer: We are very sorry for the imprecise description. We have made the corrections and indicated the changes using yellow highlights in the revised manuscript.

Lane 99: There is no information found about the immunoblot assays, particularly the antibodies used.

Answer: Thanks for the valuable suggestions, we have described more detailed information of immunoblot assays in the “Methods” section in the revised manuscript as below:

“The following antibodies were used for immunoblot analysis: Anti-STING (1:1000, #13647), anti-TBK1 (1:1000, #3504), anti-p-IRF3 (1:1000, #4947), anti-IRF3 (1:1000, #4302), anti-p-STAT1 (1:1000, #9167), anti-cGAS (1:1000, #31659), and anti-streptavidin-HRP (1:1000, #3999) were purchased from Cell Signaling Technology. Anti-NMT1 (1:2000, ab186123), anti-LC3B (1:2000, ab192890), and anti-p-TBK1 (1:1000, ab109272) were purchased from Abcam. Anti-Myc (1:5000, M4439) and anti-Flag (1:5000, F1804) antibodies were purchased from Sigma. Anti- β -actin (1:20000, 66009-I-Ig) and anti-ARF1(1:1000, 20226-1-AP) were purchased from Proteintech. Anti-HA (1:2000, CB051) was purchased from Origene.”.

3) The authors ping-pong between experiments in PMs with endogenous proteins and analysis in HEK293T cells transiently transfected with different plasmids without providing details nowhere including M&M, once again making it difficult to appreciate the appropriateness of

the results' conclusions. The authors should explain why these types of experiments in HEK293T cell with overexpressed proteins instead to make the experiments in MPs and looking at the endogenous proteins.

Answer: We are sorry for the unclear description, and all the conclusions obtained from the HEK293T cells with overexpression of the plasmids were verified in PMs. Furthermore, MEFs were used for immunofluorescence analysis because of their large size and relatively isolated organelles. We have added cell types other than PMs to all panels in the revised manuscript.

This is the case, for example, of Fig. 3a, whose insufficient legend and shown results do not help to understand the authors' conclusion. In some cases, as in Fig 4c, I do not know if experiments have been done in HEK cells or MPs. The authors need to better clarify all these points.

Answer: We accepted the valuable suggestion, we have added the cell types other than PMs to all panels in the revised manuscript, such as Fig. 3a, 4b, 4e, 7a, 7c, 7f, and 7g.

4) Quantification of all immunoblot bands should be made and reported together with further information on Western blots, including the antibodies used.

Answer: We accepted the valuable suggestion. We performed analysis of grey value of all quantitative immunoblots and described more detailed information of immunoblot assays in the "Methods" section.

5) Lane 173: increase of NMT1 expression should be revealed by immunoblot using Ab against NMT1. The authors have this Ab.

Answer: Thanks for the valuable suggestions and we performed the additional experiments accordingly. NMT1 was upregulated during HSV-1 infection (Fig.6b). We added these new data in the revised manuscript.

Figure 6a

Figure 6b

6) Line 189: why 2 bands for Arf1 in Fig. 7b but not in all other panels?

Answer: The difference may be due to the different buffer solutions used for Click chemistry in PMs in Fig. 7b, but not in Fig. 7d, resulting in the part of endogenous ARF1 protein with different migration distances during the SDS-PAGE.

7) What can we say about the ratio between Myristoylated ARF1 and the total amount of ARF1?

Answer: Due to the limitations of the current technology, only labeled myristate acid can be used to stimulate cells and detect the myristate changes of target proteins, so it is difficult to quantify the dynamic myristoylation changes during HSV-1 infection. Our results showed that N-myristoylated ARF1 was also detected during HSV-1 infection, suggesting exogenous myristic acid could induce the myristoylation of ARF1.

8) Line 132: where are the experiments with MG132?

Answer: We are sorry for the mistake, we added the references of Fig. 4b in the revised manuscript. Treatment with 3-methyladenine (3-MA) or chloroquine (inhibitors of the autophagy-lysosome degradation pathway), but not the proteasome inhibitor MG132, reversed the myristic acid-induced STING protein decrease in cGAMP-activated HEK293T cells (Fig. 4b).

9) Fig. 4c: why does STING migrate differently depending on the conditions? And more

particularly why detection in Fig. 4c is not similar to that in Fig 4a and Fig. 4c and 4d – chloroquine and + Myristic acid are not equivalent.....?

Answer: Thanks for the valuable suggestions. The migrate differently of STING may be due to that acrylamide gel was prepared without well-mixed, and we re-performed the experiments in Fig. 4c. Chloroquine treatment also reversed the inhibitory effect of myristic acid on STING, TBK1, and IRF3 protein expression in HSV-1-infected PMs.

Figure 4c

10) Line 142: specify that it is after cGAMP and HSV-1 stimulation.

Answer: Thanks for the valuable suggestions, and we have corrected it as “Myristic acid treatment promoted HSV-1 infection-induced STING-LC3 puncta formation and cGAMP-induced LC3-I to LC3-II conversion”.

11) A current model of how TBK1 and IRF3 start IFN1 expression would have helped to better understand what it is known in the literature and what the authors bring as new knowledge.

Answer: Thanks for the valuable suggestions. To help to better understand what our study brings as new knowledge, we marked the results confirmed in our study with **green arrows**, while the known process of how STING activation of TBK1 and IRF3 is marked with black arrows (Fig. S6).

Figure S6

Answer: Thanks for the valuable suggestions and we have added a list of abbreviations as “Supplementary Table 2” in the revised manuscript.

Reviewer #3:

In this well-written manuscript, the authors provide intriguing evidence that the long-chain fatty acid, myristic acid, might act as a metabolic checkpoint that regulates the balance between STING-dependent type I interferon responses and the ability of this innate immune signaling component to induce autophagy in elicited murine macrophages. The present studies are interesting and provide insights into the mechanisms by which potentially detrimental immune responses to viral challenge are regulated. The mechanistic investigations described are generally thorough and convincing, but the present study suffers from limitations related to the selection of elicited peritoneal macrophages for study and inadequately addressed virological aspects.

Answer: We appreciate your precious time in reviewing our manuscript. In accordance with your valuable suggestions and comments, we carefully revised the manuscript, and several additional experiments were performed. In doing so, we have strengthened mechanistic details and the physiological relevance of our findings.

MAJOR POINTS

1. The reason for the selection of elicited peritoneal macrophages in these studies is unclear since other immune cell types such as dendritic cells are more important sources of type I interferon production. Also, no attempt is made to verify the present findings in bone marrow-derived macrophages or human myeloid cells.

Answer: Thanks for the valuable suggestions. Macrophages are important mediators of host defense and sources of type I interferon production, thus we investigated the role of SFA in macrophages during HSV-1 infection. Furthermore, we performed additional experiments according to your suggestions. Similarly, myristic acid treatment inhibited HSV-1- and ISD-induced *Ifnb* expression in BMDMs and THP1 cells (Fig. 1i and 1j). These results confirm the function of myristic acid in different types of macrophages. We added these new data in the revised manuscript.

2. It is stated that no other short chain fatty acids have an effect, but c16:1 looks close to being so in Figure 1A but fails to be significant perhaps due to the limited sample size. Similarly, palmitic acid and linoleic acid show 15-25% reductions in HSV-1 infection-induced *Ifnb1* mRNA expression in Figures 1B and S1A. It should also be noted that no other SFAs were studied further and so the use of the term “selectively” cannot be used.

Answer: Thanks for the valuable suggestions. we have re-performed the experiment and increased the sample size. The results showed that myristic acid treatment inhibited HSV-1 infection-induced IFN-β production, and we deleted the word “selectively” and corrected as “Myristic acid most significantly attenuated HSV-1 infection-induced *Ifnb1* mRNA expression in these major SFAs, including palmitic acid, linoleic acid, and stearic acid (Fig. S1a).”

expression in Figure S2 but significant differences are shown in STING and Irf3 mRNA levels.

Answer: These results may be caused by the limited sample size, so we re-performed the experiment and increased the sample size. The results showed that myristic acid treatment did not affect the mRNA expression of *Irf3*, *Tbk1*, and *Sting* in HSV-1-infected PMs (Fig. S2).

Figure S2

Furthermore, the discrepancy of p-TBK1 may be due to the difference in total protein amount between the different groups, and we re-performed the experiment in Figure 1f, the results showed that myristic acid treatment did not suppress SeV infection-induced phosphorylation of TBK1, IRF3, and STAT1 (Fig. 1f).

Figure 1f

3. Figure 1A shows considerable variability suggesting an insufficient sample size and the p-TBK1 blot is of poor quality in Figure S1c. The myristic acid-induced decreases in p-IRF3 in Figs 3F and G, and p-TBK1 in Fig. 3F, are not convincing. Also, please provide quantification of protein expression for Figures 5i and S4c, and quantification is lacking for the effects in Figure 7e.

Answer: We accepted the valuable suggestion. Although the sample size was only 4 cases, the samples were obtained from 4 independent biological individuals. Myristate acid was the only

SFA with statistical difference and the serum myristic acid from all 4 mice was decreased respectively. The discrepancy of these phosphorylated protein may be due to the difference in total protein amount between the different groups, and we re-performed these experiments with analysis of grey value. These results showed that myristic acid treatment suppressed ISD- (Fig.S1c), cGAMP- (Fig.3f), and DMXAA- (Fig.3g) induced phosphorylation of TBK1, IRF3, and STAT1, but not SeV (Fig.1f) infection.

Answer: Thanks for the valuable suggestions. The readout times of different stimulants was determined based on previous studies (*Nat Immunol.* 2020;21(7):727-735.) and the expression pattern of p-IRF3 (A and B), and readout times were chosen to optimally activate the innate immune response. For example, phosphorylation of IRF3 was optimally induced by HSV-1 infection for 4 hours, whereas SEV infection for 8 hours. Furthermore, the number of experimental repeats were added in the revised manuscript.

5. In Figure 2, the descriptions of the *in vivo* treatments and HSV-challenge are not sufficient, and the viral strains used in these studies and their validation are not described. The study of the effect of treatments on HSV are limited to assessments of UL30 mRNA expression. It is unclear why HSV burden and leukocyte recruitment was limited to lung tissue.

Answer: Thanks for the important comments. We have described the details of the HSV-1 challenge *in vivo* in the part of “Methods”. Herpes simplex virus type 1 (HSV-1, Kos strain) was used, which has been verified after acquisition and stored subpackage in -80°C. Detection of HSV-1 UL30 mRNA expression by qPCR is widely used to evaluate HSV-1 virus replication in tissues (*Nat Immunol.* 2020;21(7):727-735.; *Nat. Commun.* 2017; 8:15534.; *Cell Mol. Immunol.* 2018;15(10):907-916.). We further examined the replication of HSV-1 in other tissues by qPCR including the brain and spleen (Fig. 2e). The replication of HSV-1 in the spleen, brain, and lung was enhanced in myristic acid-treated mice than in control mice at 72 hours after viral infection as monitored by the expression of the HSV-1 UL30 mRNA (Fig. 2e). HSV-1 infection can rapidly induce inflammatory cell infiltration and cause injury in lung tissue (*Am J Respir Crit Care Med.* 2007;175(9):865-6), therefore HE staining was widely used to detect lung lesions (*Cell Res.* 2016;26(12):1302-1319; *Nat Commun.* 2017; 8:15534.). We added the new data in the revised manuscript.

6. The authors have a tendency to overstate their results, e.g. on lines 164 and 176, effects are described as be “reversed” but are only partially so.

Answer: Thanks for the valuable suggestions, and we have corrected it as “Myristic acid-mediated decrease in IFN- β expression could be partially reversed by DDD85646 or IMP-1088 treatment (Fig. 5f–h).” and “*Nmt1* knockdown partially reversed the decrease in phosphorylation of TBK1 and IRF3 and *Ifnb1* mRNA expression in myristic acid-treated PMs (Fig. 6g and 6h).”.

Similarly, the approximately 30% inhibition by NMT1 is described on line 174 as being “markedly”.

Answer: We accepted the valuable suggestion and deleted “markedly” as “NMT1 inhibited cGAS-STING-triggered IFN- β reporter activation (Fig. 6c).”.

In lines 235-242, please indicate that this is hypothesized from the present study.

Answer: We accepted the valuable suggestion, and we have corrected it as “Viruses infection downregulate cellular level of myristic acid to release the self-limitation of STING-dependent IFN responses by suppressing autophagic degradation of STING complex. As a result, the reduction of myristic acid in macrophages caused by viral infection promotes viral clearance by enhancing STING-dependent IFN responses. Therefore, myristic acid may serve as a metabolic checkpoint that determine the consequences of the dynamic interaction between invading viruses and host immune cells.”.

Also, the notion that myristic acid could ameliorate autoimmune disorders (Lines 254-256) is highly speculative and is not supported by the present study.

Answer: We accepted the valuable suggestion and we have corrected it as “myristic acid is speculative to suppress excessive STING activation and thus ameliorate related autoimmune disorders as a dietary supplement”.

Finally, the descriptive title of Figure S1 is inaccurate as the data provided does not definitively show inhibition of the cGAS pathway, and in line 115, cGAS dependence was not

proved.

Answer: We accepted the valuable suggestion and we have corrected the title to “Myristic acid suppresses cGAS-dependent antiviral immune responses.”.

MINOR POINTS

1. The abstract does not denote the species and cell types studied.

Answer: We accepted the valuable suggestion, and we added such information in the revised manuscript.

2. Figure 1C is incorrectly referenced as Figure 1B on line 86.

Answer: We are very sorry for this mistake, and we have corrected it in the revised manuscript.

3. In Figure 4B, protein expression is labeled as Myc when STING appears to be intended.

Answer: We accepted the valuable suggestion, and we have corrected it as “STING-Myc” in the revised manuscript.

4. In line 228, please expound on why such an immune hub is “advantageous”.

Answer: We accepted the valuable suggestion, and we have further described this section as “Pathogen infection extensively remodels host lipid metabolism to enable the transfer of fatty acids from lipid droplets, which serve as an innate immune hub that is advantageous for organizing an intracellular host defense by accumulating and using immune proteins.”.

5. Please address the following grammatical/syntactical errors:

Line 92 change to “infection- and ISD-induced expression”

Line 99 change to “but not SeV, induced phosphorylation”

Line 206 change to “triggered autophagy induction and IFN production are”

Line 211 change to “proteins”

Line 215 change to “we determined that the endogenous SFA myristic acid facilitates”

Line 231 change to “Nitro-fatty acids are endogenously formed in”

Line 671, change “extended” to “supplemental”

Answer: Thank you very much for the reviewer's careful review of our article. We have corrected the mistakes in the revised manuscript.

REVIEWERS' COMMENTS

Reviewer #1 (Remarks to the Author):

The authors have done a great job improving the work (both with data and modification of the conclusions to match the data), and I now find good alignment between the data and the conclusions.

Regarding the new data in Fig 4f, I find it important to have this quantified.

Reviewer #2 (Remarks to the Author):

The authors have improved their manuscript on the STING-dependent autophagy and IFN responses via N-myristoylation of ARF1, according to the many concerns previously raised by myself and other reviewers. The authors performed several additional experiments strengthening the highlighted mechanistic process. Only a minor number of reviewers' concerns were not raised by the authors, which I do not think strictly necessary for publication.

Reviewer #3 (Remarks to the Author):

The present manuscript has been markedly improved by the addition of more study repeats that improve the scientific rigor of the work and extensive new experimentation that further validates the conclusions drawn. The study has been improved by the addition of experiments featuring bone marrow-derived macrophages and the THP1 human monocytic cell line. More information is provided regarding the virus employed and viral mRNA expression levels are now assessed in brain and spleen in addition to the lung. Immunoblot expression levels have been quantified and the authors are now more circumspect with regard to the description of the data and their conclusions. The considerable variability seen in Figure 1A continues to be somewhat worrying and the manuscript contains some syntactical/grammatical errors, but these concerns are relatively minor

Reviewer #1:

The authors have done a great job improving the work (both with data and modification of the conclusions to match the data), and I now find good alignment between the data and the conclusions. Regarding the new data in Fig 4f, I find it important to have this quantified.

Answer: Thank you very much for your valuable comments and suggestions on our manuscript. Following the comments, we quantified the intensity profiles of Fig. 4f by ImageJ software, and the results showed that myristic acid treatment inhibited cGAMP-induced localization of STING at the Golgi. Thanks again for your efforts in reviewing our manuscript and hope that the correction will meet with approval.

Fig. 4f

Reviewer #2:

The authors have improved their manuscript on the STING-dependent autophagy and IFN responses via N-myristoylation of ARF1, according to the many concerns previously raised by myself and other reviewers. The authors performed several additional experiments strengthening the highlighted mechanistic process. Only a minor number of reviewers' concerns were not raised by the authors, which I do not think strictly necessary for publication.

Answer: Thanks again for your efforts in reviewing our manuscript and hope that the correction will meet with approval.

Reviewer #3:

The present manuscript has been markedly improved by the addition of more study repeats that improve the scientific rigor of the work and extensive new experimentation that further validates the conclusions drawn. The study has been improved by the addition of experiments featuring bone marrow-derived macrophages and the THP1 human monocytic cell line. More information is provided regarding the virus employed and viral mRNA expression levels are now assessed in brain and spleen in addition to the lung. Immunoblot expression levels have been quantified and the authors are now more circumspect with regard to the description of the data and their conclusions. The considerable variability seen in Figure 1A continues to be somewhat worrying and the manuscript contains some syntactical/grammatical errors, but these concerns are relatively minor

Answer: We appreciate your precious time in reviewing our manuscript. According to the comments and suggestions, several grammar and language errors were corrected in the revised manuscript. Thanks again for your efforts in reviewing our manuscript and hope that the correction will meet with approval.